# Fold localisation at pre-existing normal faults: Field observations and analogue modelling of the Achental structure, Northern Calcareous Alps, Austria

Willemijn Sarah Maria Theresia van Kooten[1,2], Hugo Ortner[1], Ernst Willingshofer[3], Dimitrios Sokoutis[3,4], Alfred Gruber[5], Thomas Sausgruber[6]

[1]Institut für Geologie, Universität Innsbruck, Innsbruck, 6020, Austria
[2]Department of Digital Business & Software Engineering, MCI - The Entrepreneurial School, Innsbruck, 6020, Austria
[3]Department of Earth Sciences, Utrecht University, Utrecht, 3584 CB, Netherlands
[4]Department of Geosciences, University of Oslo, Oslo, Norway
[5]Geologische Bundesanstalt für Österreich (GBA), Wien, 1030, Austria
[6]die.wildbach, Wilhelm-Greil-Straße 9 6020 Innsbruck, Austria

*Correspondence to*: Willemijn S.M.T. van Kooten (willemijn.vankooten@mci.edu), ORCID: 0000-0002-9784-444X

## Abstract

Within the Northern Calcareous Alps fold-and-thrust belt of the Eastern Alps, multiple pre-shortening deformation phases have contributed to the structural grain that controlled localisation of deformation at later stages. In particular, Jurassic rifting and opening of the Alpine Tethys led to the formation of extensional basins at the northern margin of the Apulian plate. Subsequent Cretaceous shortening within the Northern Calcareous Alps produced the enigmatic Achental structure, which forms a sigmoidal transition zone between two E-W striking major synclines. One of the major complexities of the Achental structure is that all structural elements are oblique to the Cretaceous direction of shortening. Its sigmoidal form was, therefore, proposed to be a result of forced folding at the boundaries of the Jurassic Achental basin. This study analyses the structural evolution of the Achental structure through integrating field observations with crustal-scale physical analogue modelling, to elucidate the influence of pre-existing crustal heterogeneities on oblique basin inversion. From brittle-ductile models, we infer that oblique shortening of pre-existing extensional faults can lead to the localisation of deformation at the pre-existing structure and predicts thrust and fold structures that are consistent with field observations. Prerequisites are a weak basal décollement that is offset by an existing normal fault and the presence of topography in the hinterland during thin-skinned deformation. Consequently, the Achental low-angle thrust and sigmoidal fold train was able to localise at the former Jurassic basin margin, with a vergence opposite to the controlling normal fault, creating the characteristic sigmoidal morphology during a single phase of NW-directed shortening.

## 1 Introduction

The deformational style and tectonic history of Earth's orogenic belts is strongly influenced by pre-existing structural elements. In particular, structures formed by shortening and inversion of pre-orogenic basins are often governed by the reactivation of former basin-bounding normal faults as summarised by Turner and Williams (2004) or Cooper and Warren (2020). Consequently, previous studies have focused on the mechanics of fault reactivation (e.g., Etheridge, 1986; Nielsen & Hansen, 2000; Sibson, 1995; Tong & Yin, 2011), or present regional case studies of basin inversion (e.g., Héja et al., 2022; Kley et al., 2005; Thorwart et al., 2021). Existing literature presenting analogue and numerical modelling of inversional settings is extensive and includes the reactivation of planar and listric extensional fault systems of various orientation as well as basin inversion orthogonal or oblique to the basin axes (e.g., Amilibia et al., 2005; Bonini, 1998; Brun & Nalpas, 1996; Buchanan & McClay, 1992; S. J. H. Buiter & Pfiffner, 2003; Dubois et al., 2002; Konstantinovskaya et al., 2007; Koopman et al., 1987; Sassi et al., 1993; Zwaan et al., 2022). In most models, newly formed reverse faults are synthetic to the pre-existing normal

faults (see also Bonini et al., 2012; their Figure 3), whereas the formation of thrust faults antithetic to the former basin-bounding faults is seldomly described.

In this study, we use an example from the Northern Calcareous Alps (NCA) in Austria (Figure 1), in which basin inversion during Alpine orogeny has created a low-angle thrust antithetic to a normal fault bounding a Jurassic extensional basin. In the hanging wall of this thrust, a fold train outlines the margins of this basin, forming a characteristic sigmoidal shape that has

been the subject of geological investigations since the beginning of the 20th century (Ampferer, 1902, 1941; Fuchs, 1944; Nagel, 1975; Nagel et al., 1976; Quenstedt, 1933). We use the example of the Achental structure and its underlying basin geometry as a starting point for analogue modelling, to simulate the oblique inversion of a former extensional basin and the deformation of a brittle-ductile sedimentary succession at its margins. The main aim of this study is to better understand strain localization and associated deformation by folding and thrusting at oblique basin bounding faults. In particular, this study

examines whether a sigmoidal hanging wall shape outlining former basin margins can be explained by a single phase of oblique shortening and if so what are the critical rheological and kinematic parameters to obtain such structures. Our modelling results are then compared to the Achental structure, to better understand the formation of complex inversion structures in fold-and-thrust belts.

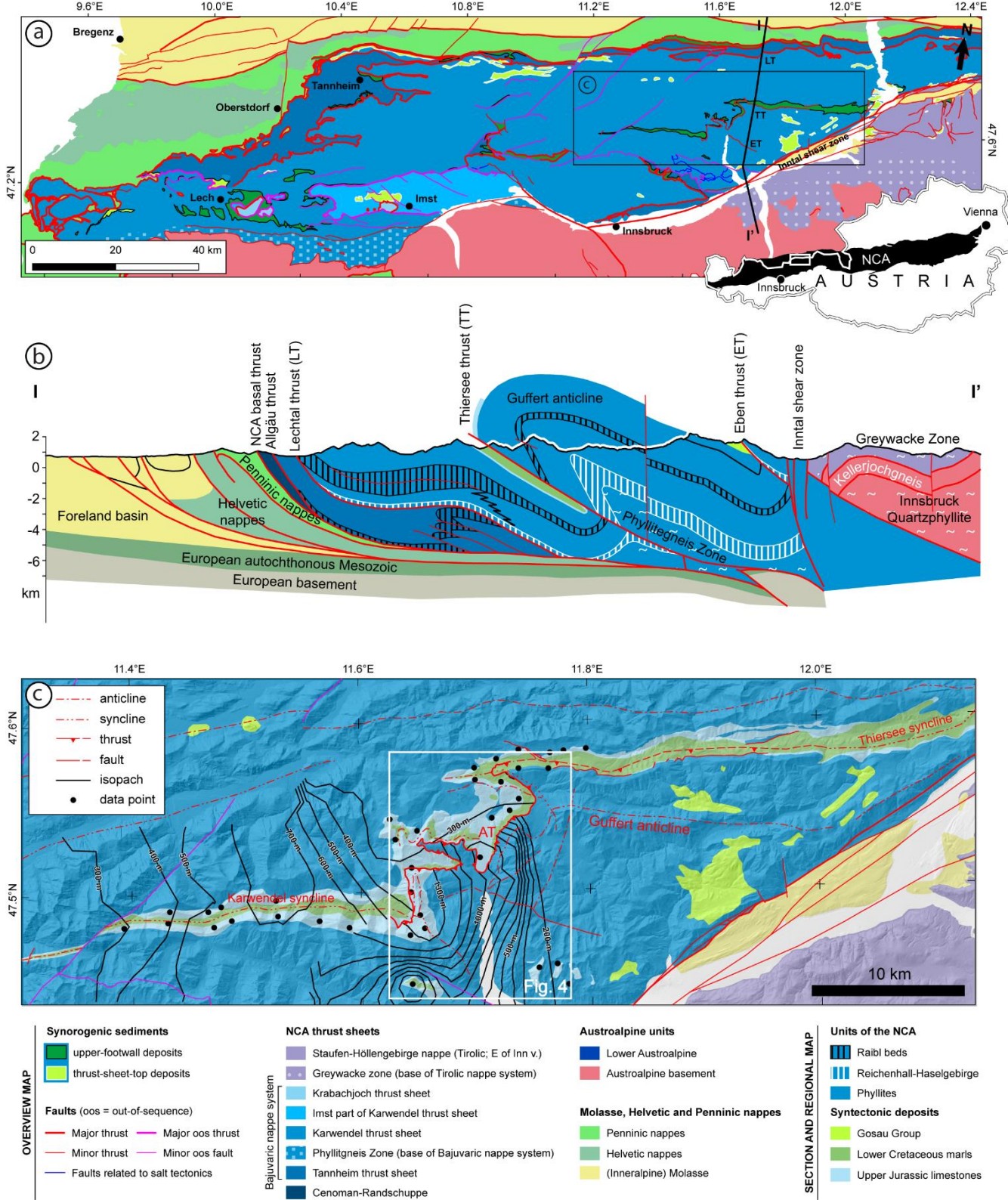

**Figure 1:** (a) Geological map of the Achental structure within the Karwendel thrust sheet. Top inset shows the location of the NCA in Austria. Rectangle shows location of Figure 1c. Modified after Ortner and Kilian (2022). (b) N-S cross-section of the NCA at ~ 11.7°E, crossing the Achental structure. (c) Isopach map of the Upper Jurassic Oberalm Fm, with data compiled from Nagel et al. (1976) and K.-I. Schütz (1979). The hanging wall and footwall of the Achental thrust (AT) were contoured separately, single outliers in the data were ignored. Major thickness changes occur from W to E, a maximum is observed in the southern hanging wall of the thrust.

## 2 Geological setting

The Northern Calcareous Alps (NCA) of Austria have a polyphase tectonic and sedimentary history that started in the Permian.

The following evolutionary stages are distinguished (Figure 2):

1) Deposition of a Permian-Triassic carbonate platform succession on the south(east)ern passive margin of Pangaea bordering the Neotethys from the Permian to the end of the Triassic (e.g., Haas et al., 1995; Lein, 1987; Schmid et al., 2004; Schmid et al., 2008; Stampfli et al., 1998).

2) Early to Middle Jurassic rifting and subsequent opening of the Alpine Tethys that separated the Apulian/Adriatic microplate from the European continent lead to subsidence and drowning of the Triassic carbonate platforms (e.g., Faupl & Wagreich, 1999; Froitzheim & Manatschal, 1996; Schmid et al., 2004). Rift-related faulting established the normal faults that were inverted during Cretaceous orogeny (Eberli et al., 1993; Ortner et al., 2008).

3) Shortening related to Late Jurassic obduction of Neotethys oceanic crust onto the southeastern Apulian margin heralded inversion of this margin that culminated in the Cretaceous orogeny (Schmid et al., 2004; Stüwe & Schuster, 2010). During the Cretaceous orogeny the NCA were a typical thin-skinned fold-and-thrust belt at the external margin of the Austroalpine orogenic wedge in lower plate position (Eisbacher & Brandner, 1996; Ortner & Kilian, 2022).

4) During the Late Cretaceous this wedge was transported toward the northwestern passive margin of the Apulian plate, which became an active margin through subduction of the Alpine Tethys (Ortner & Sieberer, 2022; Stüwe & Schuster, 2010; Willingshofer et al., 1999).

5) Late Eocene closure of the Alpine Tethys caused collision between the lower, European and upper, Apulian plates and thus a second, Paleogene phase of mountain building, referred to as "Alpine orogeny" (Eisbacher & Brandner, 1996; Ortner, 2003b; Schmid et al., 2004; Stüwe & Schuster, 2010). In the course of this process the Austroalpine units and the NCA fold-and-thrust belt were transported piggyback onto the European margin. Contrasting with Cretaceous orogeny, the Austroalpine wedge formed the upper plate during Paleogene orogeny.

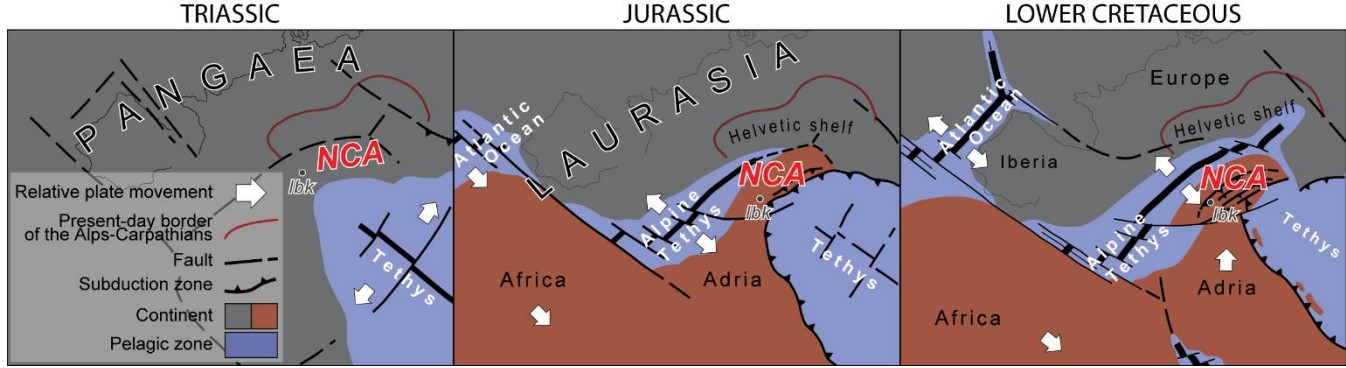

**Figure 2:** Developmental stages in the tectonic evolution of the Northern Calcareous Alps, modified after Schuster et al. (2019). (a) Permotriassic stage with sedimentation on the passive margin of Pangaea. (b) Early-Middle Jurassic opening of the Alpine Tethys, with sedimentation in basins and on swells. (c) Cretaceous orogeny. (d) Late Cretaceous subduction leading to Late Eocene closure of the Alpine Tethys and Alpine orogeny. The present-day location of Innsbruck (Ibk) is marked.

Relevant for this study are the Jurassic rifting that created sedimentary basins and the Cretaceous and Paleogene shortening that led to basin inversion. Jurassic extensional basins formed along intersecting fault systems with N-S trending normal faults and E-W trending transform faults, as documented in the Eastern Alps of Switzerland (Eberli, 1985, 1987; Weissert & Bernoulli, 1985). Similar orientations of extensional normal faults have been found in the Western Alps (Lemoine et al., 1986). The existence of an extensional basin in the Achensee region is indicated by filled fissures that occur in Oberrhätkalk (Gruber et al., 2022), strong lateral thickness variations of (Upper) Jurassic strata (Figure 1b) (Nagel et al., 1976; K.-I. Schütz, 1979) and characteristic breccias that are locally intercalated in basinal Jurassic sediments (Brandner et al., 2011; Channell et al., 1990; Channell et al., 1992; Spieler & Brandner, 1989). Cretaceous to Paleogene shortening within the NCA is characterized by two separate directions of transport. During Cretaceous orogeny the thrust sheets of the NCA were transported and stacked in WNW- to NW-direction (Eisbacher & Brandner, 1996), whereas Paleogene shortening of the NCA fold-and-thrust belt was associated with N- to NNE transport directions (Schmid et al., 1996). These differences are well documented within the

basement units of the Austroalpine wedge (Froitzheim et al., 1994), althoughsynorogenic growth strata show continuous shortening within the NCA (Ortner, 2001, 2003a; Ortner et al., 2016; Ortner & Gaupp, 2007).

## 2.1 Sedimentary succession

The sedimentary succession of the NCA encompasses a late Permian to Oligocene stratigraphy. The Permotriassic succession reflects subsidence of the south(east)ern passive margin of Pangaea from continental to shallow marine conditions. During the Permian-Middle Triassic, the marine ingression of the Tethys led to the formation of a subtidal-supratidal marine environment, where evaporitic Haselgebirge-Reichenhall succession accumulated. The Haselgebirge contains large amounts of halite, gypsum and anhydrite in a shale matrix (Leitner & Neubauer, 2011). The thickness of Haselgebirge is highly variable, due to salt tectonics and solution, but a primary thickness of 500-1000 m can be expected (Spötl, 1989: Fig. 2). Lower Triassic sandstones (Alpine Buntsandstein) overlie the Haselgebirge and are followed by an Anisian limestone-dolostone-evaporite succession (Reichenhall Formation (Fm; Figure 3) with a thickness of 80-200 m in the Achensee region (Nagel et al., 1976). The Haselgebirge-Reichenhall succession represents a rheologically incompetent part of the stratigraphic column, forming the principal décollement of the NCA (Eisbacher & Brandner, 1995). The Eben thrust (Figure 1) brings the Haselgebirge-Reichenhall succession to the surface in the Achensee region. Its presence in the subsurface has been inferred from depth-extrapolated cross sections based on TRANSALP and industry reflection seismic lines (Auer & Eisbacher, 2003: their Figure 15).

The evaporitic Haselgebirge-Reichenhall succession transitions into a succession of Anisian-Ladinian limestones (Alpine Muschelkalk Group; Bechstädt & Mostler, 1974). These vary from shallow-water, strongly bioturbated mud- to wackestones (e.g., Virgloria Fm, Steinalm Fm) to basin marginal and basinal limestones (Reifling Fm; Rüffer & Zühlke, 1995). The uppermost Alpine Muschelkalk Group interfingers with the Wetterstein limestone (Anisian–Carnian), which forms the first of two major carbonate platforms of the NCA (Figure 3). Although the Alpine Muschelkalk Group and Wetterstein limestone show a general thickness between 1000 to >2200 m in the Achensee region (Gruber et al., 2022; Nagel et al., 1976; Sausgruber, 1994b), cross sections in Gruber et al. (2022) show a thickness up to 3500 m. These thickness variations might be a result of salt tectonics in the Triassic (e.g., Granado et al., 2019; Kilian et al., 2021; Ortner & Kilian, 2022) , however, minibasin margins with typical stratal geometries such as wedges or hooks (Giles & Rowan, 2012) or flaps (Rowan et al., 2016) have not been documented in the Achensee region so far. About 400 m of Carnian sandstones, shales, dolomites, evaporites and limestones (Raibl beds) overlie the Wetterstein platform unconformably (Gruber et al., 2022; Jerz, 1966; Krainer et al., 2011; Sausgruber, 1994a). Norian dolomites (Hauptdolomit carbonate platform) (Fruth & Scherreiks, 1982; Müller-Jungbluth, 1968; Zorlu, 2007) form the second major platform of the NCA, which is again 1-2 km thick (Donofrio et al., 2003)The Hauptdolomit platform subsequently drowns towards its top; a Norian succession of alternating limestones and dolostones (Plattenkalk; Gümbel, 1861) heralds the deposition of Rhaetian basinal shales (Kössen Fm), which interfinger with Upper Rhaetian platform carbonates (Oberrhät limestone; Golebiowski, 1991; Riedel, 1988) (Figure 3). The Ladinian and Norian platforms represent an approximately 3 km thick, mechanically competent unit. The mechanical significance of intervening Carnian shales and evaporites for deformation during the Cretaceous orogeny has been previously discussed (Kilian et al., 2021) and approached through analogue modelling in this contribution (see Section 3.2 and 4 below).

While rifting of the Alpine Tethys (Eberli et al., 1993) in the Jurassic caused subsidence and drowning of the Triassic carbonate platforms (Ortner et al., 2008), rotational block faulting caused by the extensional movement (Lackschewitz et al., 1991) resulted in a differentiation of basin and swell facies (Nagel et al., 1976; Spieler & Brandner, 1989). Early Jurassic red condensed limestones (e.g., Adnet Fm, Sinemurian–Toarcian; Sausgruber, 1994a; Spieler, 1994); were deposited on swells, and basinal strata (e.g., Allgäu Fm. Hettangian–Oxfordian) in subsided areas (Brandner & Gruber, 2011: Fig. 15). Basins and swells existed until the Middle-Late Jurassic (Brandner & Gruber, 2011; Ortner et al., 2008; Ortner & Kilian, 2016; Spieler &

Brandner, 1989). The continental margin reached its maximum subsidence in the Oxfordian (Ortner & Kilian, 2016), when
radiolarites were deposited (Ruhpolding radiolarite) (Bernoulli & Jenkyns, 1974; Nagel et al., 1976; Sausgruber, 1994a) . The
overlying Late Jurassic-Early Cretaceous (Kimmeridgian-Berriasian) stratigraphic succession (e.g., Oberalm and Schrambach
Fms) testifies to a shallowing depositional environment (Gawlick, 2004; Gruber et al., 2022; Gümbel, 1861; Kilian, 2013;
Lipold, 1854; Sausgruber, 1994a). The lower–middle Jurassic succession has a mean thickness of 290 m (based on cross
sections and compilations of Auer, 2001; Gruber, 2011; Ortner & Gruber, 2011; Sausgruber, 1994b; Spieler, 1995), while the
upper Jurassic to lower Cretaceous pelagic limestones (Oberalm Fm) reach a thickness of more than 1300 m and cover the
pre-existing relief (Nagel et al., 1976).

Increase influx of siliciclastic detritus, beginning in the Early Cretaceous, marks the onset of synorogenic sedimentation,
commencing with the Schrambach Fm (Berriasian-Aptian), consisting of marls and intercalated calcareous and turbiditic
sandstones (Lipold, 1854; Ortner, 2003a). In the western Thiersee syncline, its thickness is ~ 300-400 m (Ortner & Gruber,
2011; K.-I. Schütz, 1979), thinning to ~ 150 m east of Achenkirch (Ortner & Gruber, 2011). The overlying synorogenic
sediments of the Gosau Group (Late Cretaceous to Paleogene) complete the stratigraphic succession of the Achental region
(Ortner, 2003a).

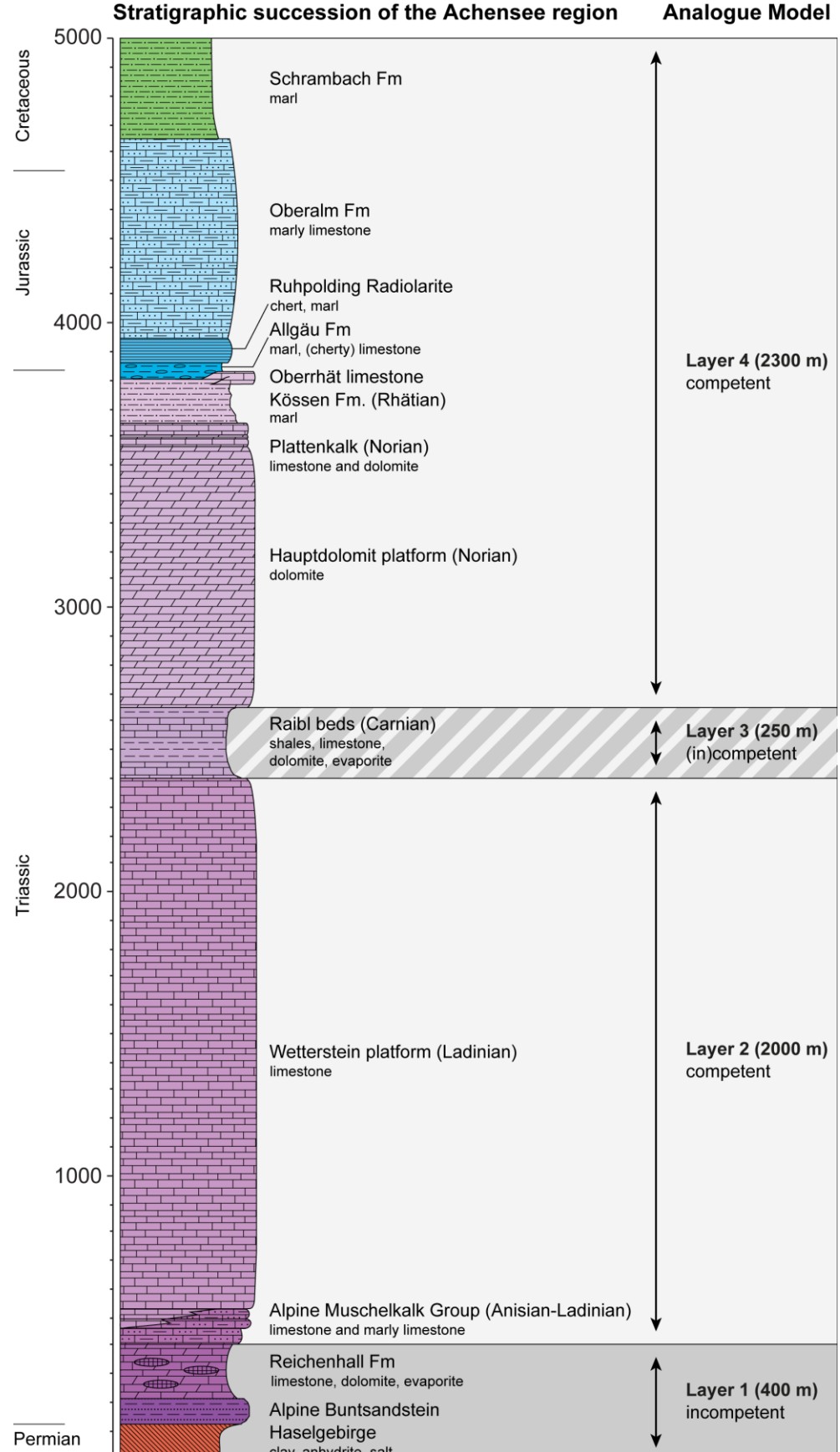

**Figure 3:** Stratigraphic overview of the Permian–Cretaceous sedimentary succession of the Karwendel thrust sheet in the Achensee area. Modified after Kilian et al. (2021).

## 2.2 Achental structure

The Achental structure is located within the Karwendel thrust sheet, following the revised nappe structure of the NCA (Kilian & Ortner, 2019; Ortner, 2016; Ortner & Kilian, 2022), and geographically surrounds the Achensee in Tyrol, Austria (Figures 1, 4). The structure is characterized by the low-angle Achental thrust, which separates the Karwendel and Thiersee synclines in its footwall, from the Guffert-Unnutz-Montscheinspitze anticline fold train in its hanging wall. It forms a structural NNE-SSW striking transfer zone between Karwendel and Thiersee synclines. The Achental thrust reaches into the cores of these folds and gradually loses offset toward the NE and SW (Ortner & Gruber, 2011). The thrust changes its orientation and stratigraphic offset throughout the Achental structure; in the north (Mahmooskopf-Natterwand section, Figure 4), the thrust strikes E-W and dips ~ 23-30° south (Ortner, 2003a and Beer, 2003, respectively) with a total displacement of ~ 5-7 km (Auer & Eisbacher, 2003 and Ortner, 2003a, respectively), based on cut-offs of Jurassic strata in Section B (Figure 5b). In the central section, the thrust strikes NNE-SSW and dips 15° SE (Spieler, 1994). The maximum stratigraphic offset of ~ 8 km (Eisbacher & Brandner, 1996) is located NE of Achenkirch, where Hauptdolomit (Norian) is thrusted onto Schrambach Fm (Early Cretaceous) (Figures 4, 5a) (Gruber et al., 2022). In the south, the Achental thrust dips SE, offsets the hinge of the Seebergspitze syncline and runs into the core of the Karwendel syncline (Figure 4). Although it separates consecutive Late-Jurassic-Early Cretaceous strata, the contrasting orientation of the strata (E-dipping north of the Seebergspitze syncline hinge; vertical to S-dipping west of the hinge) rules out a stratigraphic contact between these formations (Ortner & Gruber, 2011). However, offset is minor and and dies out not much further to the west. A series of km-scale, E-W striking north-vergent synclines (Karwendel, Gröben, Großzemmalm, Klammbach and Thiersee synclines) and associated anticlines (Scharfreuter and Hofjoch anticlines) form the footwall of the Achental thrust (Figure 4). These folds are overridden by the Achental and Leiten thrust (Ortner & Gruber, 2011: their Figure 2).

The hanging wall of the Achental thrust ("Achentaler Schubmasse" after Quenstedt (1933)) is comprised of the connected Montscheinspitze, Unnutz and Guffert anticlines (Figures 4, 5), which change strike approximately parallel to the Achental thrust. West of the Achental structure, the Montscheinspitze anticline is connected to the Karwendel syncline (Ampferer & Heissel, 1950: their section 3). In the SW corner of the Achental structure, the northern anticline limb is folded with the Seebergspitze syncline into an overturned position (Fuchs, 1944; Ortner & Gruber, 2011). The Achental thrust emplaces the anticline limb onto the Karwendel syncline (Nagel et al., 1976). The Unnutz anticline, a recumbent fold with a SE-dipping axial surface and an increasingly SSW-plunging fold axis (From N-S 205/18, 200/27 and 187/23; Ortner & Gruber, 2011), forms the N-S striking part of the fold train. The Unnutz anticline (Figure 5a) resembles a dome, rather than a cylindrical fold (e.g., Mojsisovics, 1871; Ortner & Gruber, 2011; Sausgruber, 1994a, 1994b). NE of Achenkirch, the Unnutz anticline is folded around the Rotmöserkopf synform and continues as the E-W striking Guffert anticline (Gruber et al., 2022; Ortner & Gruber, 2011) (Figures 4, 5b). The Guffert anticline has a S-dipping axial surface, sub-horizontal fold axis with ESE-WNW strike and an overturned northern limb.

The characteristic sigmoidal shape of the Guffert-Unnutz-Montscheinspitze anticline fold train has been noticed since the beginning of the 20th century (Ampferer, 1902) and is one of the greatest complexities in the geologic interpretation of the Achental structure. Over the past century, four principal hypotheses have been developed to explain its formation (see also Ortner & Gruber, 2011; Gruber et al., 2022). These include *1)* bending of an originally E-W striking anticline due to rotational movements (e.g., Ampferer, 1921, 1941; Auer, 2001; Spengler, 1953, 1956); *2)* passive dragging of the central part of the structure, caused by a larger amount of shortening in the eastern part of the structure, compared to its western part (Nagel, 1975); *3)* polyphase deformation, in which a former W-directed thrust (Achental thrust) was reactivated by N-directed thrusting (Channell et al., 1990; Channell et al., 1992; Fuchs, 1944; Ortner, 2003a; Spieler & Brandner, 1989), and *4)* forced folding of the Guffert-Unnutz-Montscheinspitze fold train at the border of a carbonate platform, or along an inverted Jurassic normal fault (Eisbacher & Brandner, 1995, 1996; Ortner & Gruber, 2011).

The kinematic history of the Achental structure is characterised by polyphase shortening (Fuchs, 1944; Ortner, 2003a; Ortner & Gruber, 2011; Spieler & Brandner, 1989). Barremian sediments below the Achental thrust indicate an Early Cretaceous maximum age of thrusting. The oldest contractional deformation in the area is indicated by an angular unconformity in both limbs of the Guffert anticline at the base of the Gosau Group (Ortner & Gruber, 2011), indicating Early Cretaceous growth of the anticline. Anticlines grow on top of décollements, and we speculate that the Achental thrust was already active. Cretaceous shortening along the Achental thrust is evident from calcite slickensides showing NNW- to NW-directed movement (Eisbacher & Brandner, 1995, 1996; Sausgruber, 1994a, 1994b). This phase of shortening caused an initial uplift and tightening of the Unnutz-Achental structure and is responsible for NW-striking high-angle dextral transfer faults, which segment the Unnutz anticline (Eisbacher & Brandner, 1995, 1996). Late Cretaceous to Paleogene (80-30 Ma) NE-directed shortening superimposed existing folds and fault movement, reactivating pre-existing NW-striking transfer faults and forming NW-plunging folds and NE-striking high-angle transfer faults (Eisbacher & Brandner, 1995). As a result, three major fold orientations are recognized in the Achensee region (Sausgruber, 1994a). Folds with NE-SW striking fold axes formed during NW-directed, pre-Gosau contraction. E-W striking fold axes formed during N-directed shortening and NW-SE striking fold axes formed during NE-directed shortening. Folding of the Achental and Leiten thrust around the E-W striking Roßstand anticline, but not around the parallel Scharfreuter anticline (Sausgruber, 1994b; Ortner & Gruber, 2011: their Fig. 2) (Figure 4) suggests that the thrust is older than the general N-directed shortening (Spieler & Brandner, 1989). However, since the oldest sediments covering both the hanging wall and footwall of the northern Achental thrust belong to the Gosau Group, N-directed shortening must have occurred post-Gosau (Ortner, 2003a), post-dating the formation of the western Achental thrust. Neogene folds superimpose older folding and can be found e.g., within the Thiersee syncline (Sausgruber, 1994b). Fold interference creates intriguing dome-and-basin structures that complicate the geological interpretation of the Achental structure. Polyphase deformation of the Achental structure is visible from field evidence, showing refolded faults and fold axes (e.g., Gruber et al., 2022; Ortner & Gruber, 2011; Sausgruber, 1994b). However, it is unclear whether two phases of deformation were necessary to create the Achental structure (Gruber et al., 2022).

A link between the Jurassic basin architecture and the present-day sigmoidal form of the Achental structure has been proposed previously, suggesting that the latter is the result of forced folding (Eisbacher & Brandner, 1995, 1996; Ortner & Gruber, 2011) that depended on pre-existing extensional structures rather than the exact direction of shortening (Töchterle, 2005). Channell et al. (1990) postulated the existence of an E-W striking sinistral pull-apart basin, forming a controlling inherited Jurassic basin-and-swell topography. This is supported by field observations. Cretaceous transport directions (Ortner & Gruber, 2011; Sausgruber, 1994a, 1994b) are oblique to the axis of the Karwendel and Thiersee synclines. If deformed within a single episode of shortening, this requires pre-existing faults for localization of folding and thrusting. Furthermore, the thickness of Jurassic deposits differs in both limbs of the Karwendel and Thiersee synclines (Nagel et al., 1976) and the maximum sedimentary thickness of Upper Jurassic strata is located in the overstep area between the two synclines (Figure 1b). The strata show a clear facies differentiation and a generally increasing thickness of basinal Jurassic strata, from 100 m near Mittenwald (30 km W of the study area) to 1100 m in the Bächental (10 km NW of the study area; e.g., Gruber et al., 2022; Nagel et al., 1976; Ortner & Kilian, 2016; K.-I. Schütz, 1979; Ulrich, 1960) (Figure 1b), showing an increased subsidence toward the SE end of the Karwendel syncline. This suggests that a basin existed that was bound by normal faults at a high angle to the faults in the synclines. Since major normal faults have not been mapped east of the Achental thrust, we interpret a W-dipping blind fault controlling deposition and facies differentiation in the Jurassic. Pliensbachian and Toarcian mass-flow sediments and scarp-breccias that are found near the basin slopes (Spieler & Brandner, 1989) support this hypothesis. Based on the aforementioned, it was proposed that this Jurassic basin-and-swell topography, in the form of a N-S to NE-SW striking pull-apart basin (Ortner & Gruber, 2011; Spieler & Brandner, 1989) or negative flower structure (Sausgruber, 1994b) bordered by E-W striking sinistral

strike-slip faults, forms the foundation of the present-day Achental structure. We therefore use this basin geometry as the base

240    for our analogue models.

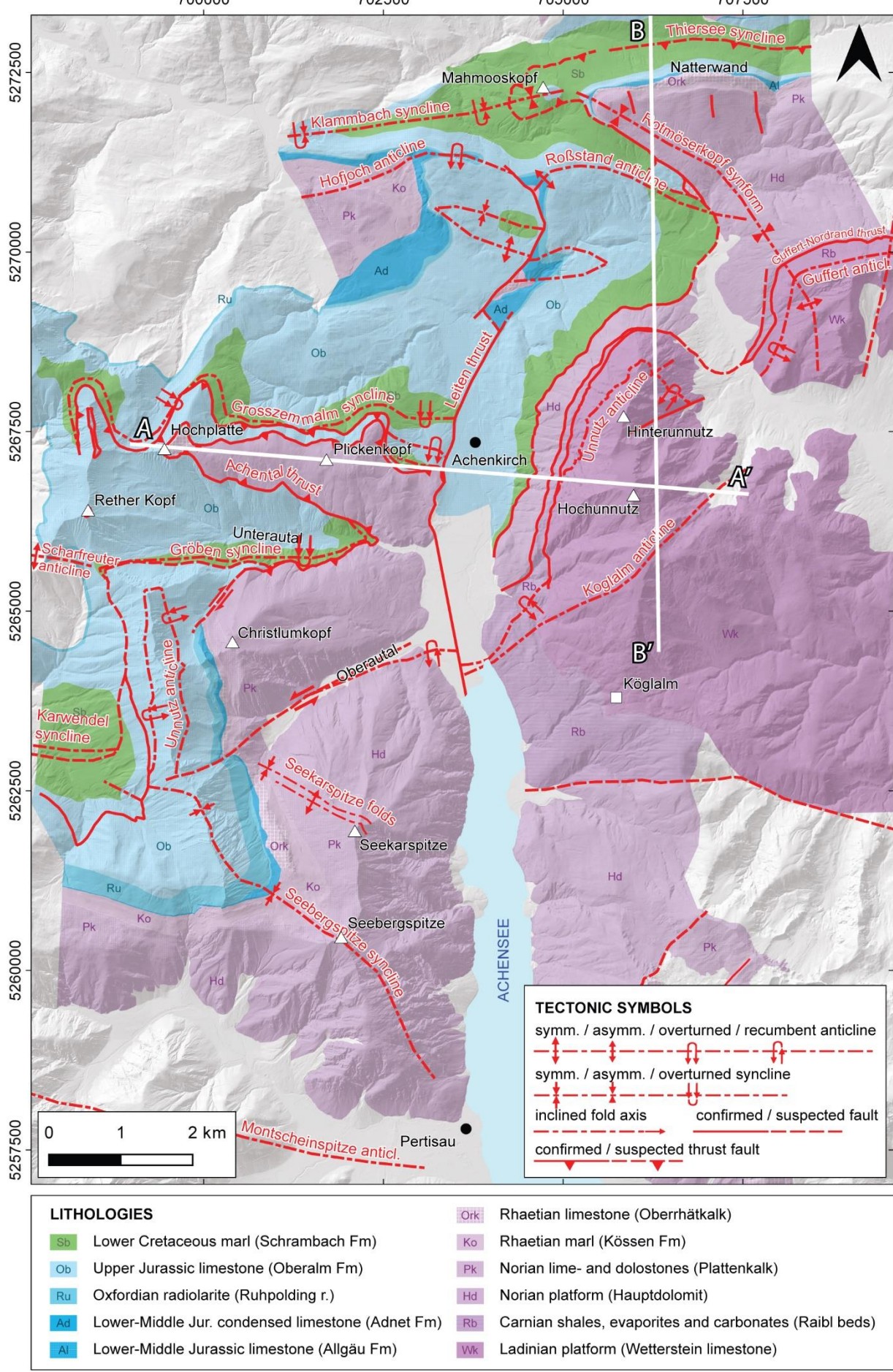

**Figure 4:** Geological map of the Achental structure compiled from Sausgruber (1994b), Spieler (1995), Auer (2001), Gruber (2011) and own data. Modified after Ortner and Gruber (2011). Sections A-A' and B-B' are shown in Figure 5.

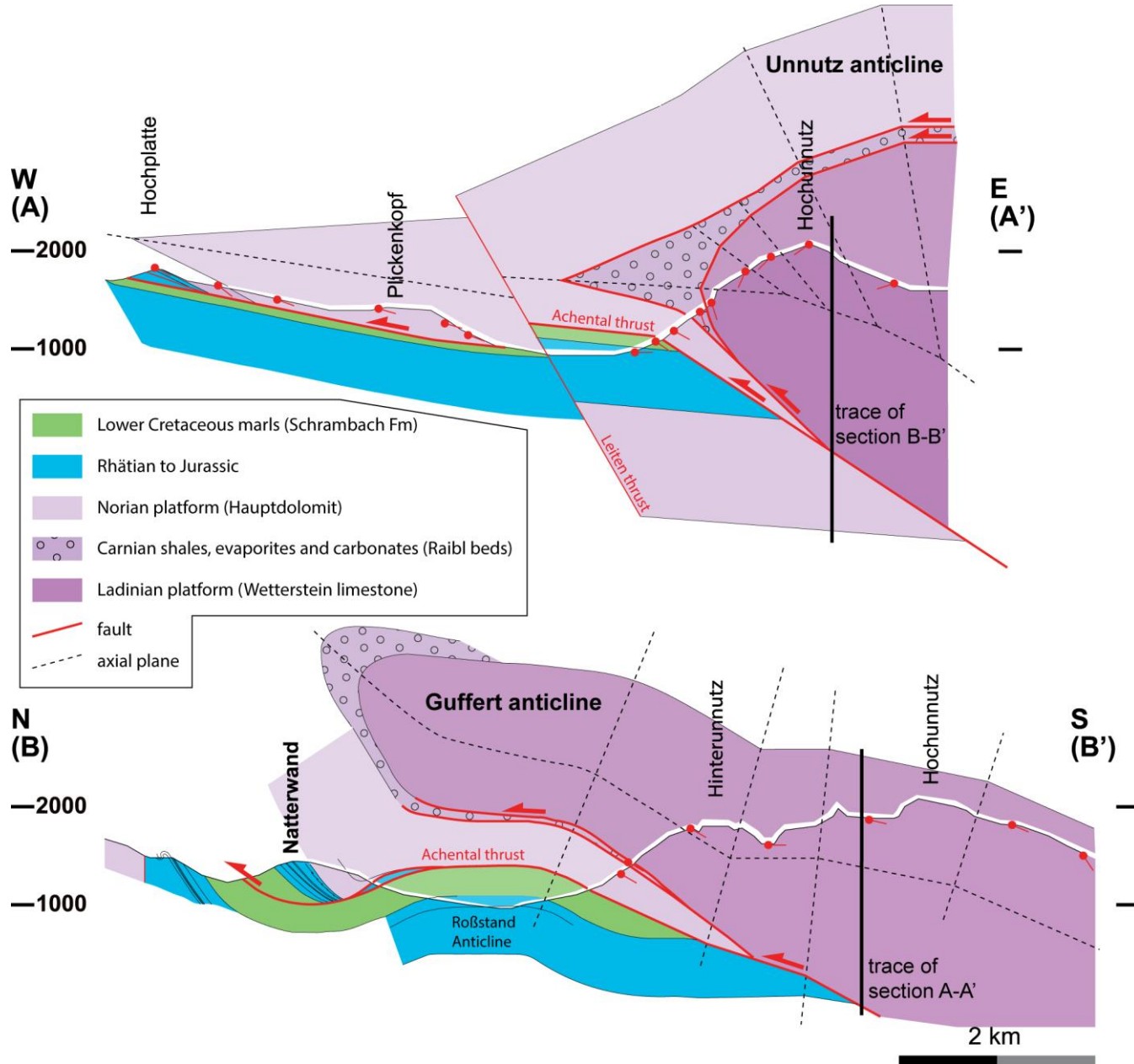

**Figure 5:** Cross sections (a) of the Unnutz anticline (Section A-A') and (b) of the Guffert anticline (Section B-B'). Section traces are marked in Figure 4. Modified after Ortner (2003a).

## 3 Analogue modelling

### 3.1 Modelling strategy

Regional-scale physical analogue modelling was conducted to investigate the influence of pre-existing basin-bounding extensional faults on oblique basin inversion, within the geological framework of the Achental structure. The goals of the experiments included testing 1) the importance of a weak basal décollement, 2) the influence of thick-skinned versus thin-skinned tectonics, and 3) the role of pre-existing structures in deformation localisation. Geological parameters from the natural example form the basis for our modelling approach. When translating geological structures into analogue models, a number of details and complexities are inevitably omitted, resulting in the creation of a simplified modelling basis that incorporates only the most important elements. In the case of the Achental structure, we have chosen to use the predicted Jurassic fault and basin geometry as a modelling basis, with a W-dipping normal fault and two E-W striking strike-slip faults. This is comparable to the basin geometry proposed by e.g., Ortner and Gruber (2011) and Spieler and Brandner (1989), and is based on facies and thickness changes within Jurassic deposits, as well as structural considerations outlined in Section 2.2. The rheological

properties of the NCA sedimentary succession, described in Section 2.1, were considered for the layering of the analogue
models. For the basal Permo-Triassic Haselgebirge-Reichenhall succession we used ductile modelling materials (layer 1;
Figures 3, 6). The middle and upper Triassic carbonate platforms are represented by brittle materials (layers 2 and 4). Rhätian
to Lower Cretaceous units are largely marls and marly limestones (Figure 3). Although these are considered less competent
units than the Ladinian and Norian platform carbonates, they are on the top of the analogue model and do therefore not control
the modelling result. The Carnian Raibl beds were modelled using brittle or ductile materials, depending on the specific model
(layer 3; Figure 6). Analogue models were set up to simulate the lithological and structural configuration of the eastern margin
of the Jurassic Achental basin, resulting in an increasing model complexity.

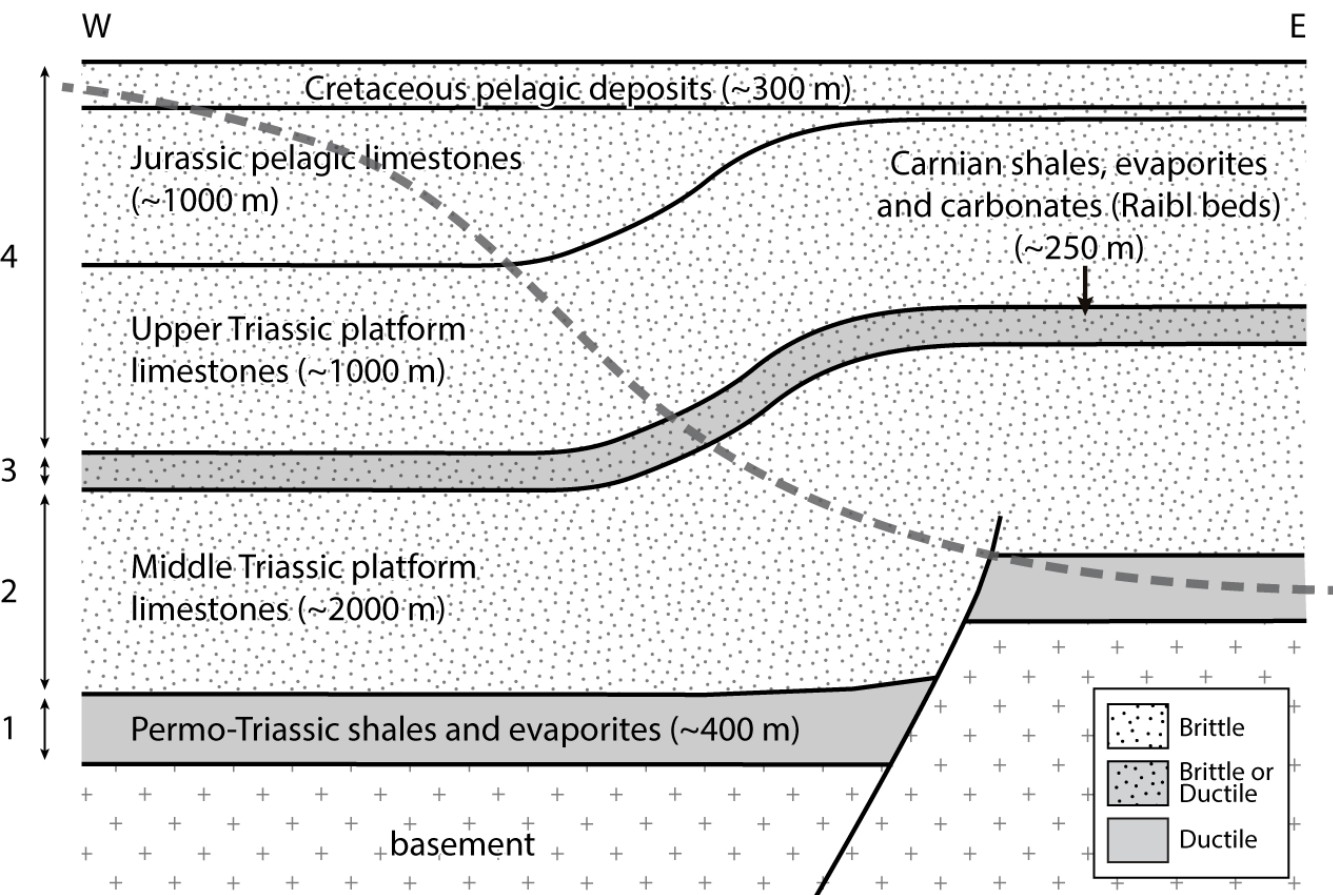

**Figure 6:** Schematic diagram showing the geological set-up of the eastern margin of the Achental basin, prior to inversion. Dashed line
represents inversional trace of the Achental thrust. Lithologies are marked brittle or ductile, depending on their mechanical properties.

**3.2 Modelling setup**

Brittle and ductile modelling materials represent different parts of the NCA and Tethyan stratigraphy (Figure 3). A ductile
layer of ~ 400 m thick, incompetent evaporites and shales, forming the basal décollement of the NCA was represented by
silicone putty in models, with a thickness of 0.4 cm (Figures 6, 7a). The silicone putty is a mixture of RBG-0910 Dow Corning
silicon polymer and iron powder (~ 32 wt.%) with a density of ~ 1360 kg m$^{-3}$. We determined a viscosity ($\eta$) of 20456 Pa s for
the modified silicone putty, using a coni-cylindrical viscometer under room temperature (21 ± 1 °C) (Lee & Warren, 1940;
Mooney & Ewart, 1934, see also Willingshofer et al., 2005). The viscosity of natural evaporites, albeit highly variable, is
estimated to be $10^{19}$ Pa s (Allen & Beaumont, 2016; Weijermars, 1986a, 1986b, 1986c; Weijermars et al., 1993; Weijermars
& Schmeling, 1986), resulting in a viscosity scale-ratio of 2.046 · $10^{-15}$. The silicone putty exhibits near-Newtonian behaviour
and has an n-value of 1.27 in laboratory tests. The predominantly carbonatic, brittle sedimentary cover of the NCA was
represented by dry quartz sand (Figure 6), a Mohr-Coulomb material, with a density of 1500 kg m$^{-3}$ (e.g., Willingshofer et al.,
2018). The quartz sand was sieved on top of the silicone putty, to create a total model thickness of 5 cm (Figure 7a). The

density of the natural prototype was approximated by the density of Triassic Muschelkalk, which is ~ 2680 kg m$^{-3}$ (Manger, 1963: p. E31), resulting in a density ratio ($\rho$*) of 0.56 between model and nature. An intermediate layer of Upper Triassic carbonatic-evaporitic strata (Raibl beds) was modelled as either brittle or ductile layer, depending on the model-specific set-up (Figure 7a-b). For a discussion whether this unit behaves incompetent or not, see Kilian et al. (2021). Assuming that inertial forces can be neglected in the analogue models (see discussion in Del Ventisette et al., 2007; Wickham, 2007), the scaling of time and length is allowed to deviate from the principle of dynamic similarity and their "ratios can be considered as independent variables" (Dombrádi et al., 2010, p. 109). The principles of dynamic, geometric and rheological scaling are discussed in Hubbert (1937), Ramberg (1981), Weijermars and Schmeling (1986), Merle and Abidi (1995), Brun (1999), Sokoutis et al. (2000; 2005) while their relationships between model and nature are summarized in Table 1.

**Table 1:** Summary of model parameters. Experimental material parameters from Willingshofer et al. (2005; 2018). For strength envelops, see Figure 7b.

| Materials/parameters | | Model | Nature | Ratio (m/n) |
|---|---|---|---|---|
| *Brittle layer* | Thickness | Max. 0.05 m | Max. 5000 m | 10$^{-5}$ |
| *(Quartz sand)* | Density | 1500 kg m$^{-3}$ | 2680 kg m$^{-3}$ | 0.56 |
| | Strength | Max. 1471.5 Pa | Max. 2.63 $\cdot$ 10$^{-8}$ Pa | 5.597 $\cdot$ 10$^{-6}$ |
| *Ductile layer* | Thickness | 0.004 m | 400 m | 10$^{-5}$ |
| *(Silicone mixture)* | Density | 1360 kg m$^{-3}$ | 2450 kg m$^{-3}$ | 0.56 |
| | Viscosity | 20456 Pa s | 10$^{19}$ Pa s | 2.046 $\cdot$ 10$^{-15}$ |
| | Strength | 28.41 Pa | 2550933 Pa | 1.11 $\cdot$ 10$^{-5}$ |
| | n-value | 1.27 | | |
| Length | | | | 10$^{-5}$ |
| Strain rate | | 2.77778 $\cdot$ 10$^{-6}$ m/s | 1.02422628 $\cdot$ 10$^{-10}$ m/s | 27137 |
| Time | | 28800–39600 s | 7.8146208 $\cdot$ 10$^{-13}$– 1.0747469 $\cdot$ 10$^{-14}$ s | 3.685 $\cdot$ 10$^{-10}$ |
| Bulk shortening | | 0.08–0.11 m | 8000–11000 m | 10$^{-5}$ |

Model materials were placed in a rectangular fault box with metal bars as sidewalls and a wooden piston of 40 or 83 cm width as a moving wall (Figure 7c). We chose a length scale-ratio (L*) of 10$^{-5}$, so that 1 cm in the model represents 1 km in nature (Table 1). In all experiments, a pre-existing normal fault, similar to the boundary fault of the Achental basin was represented by a rigid basal plate (footwall block) with a 60° dipping ramp (fault plane). In model-specific set-ups, the direction of shortening (at 90 or 45° to the ramp), the movement of the basal plate (fixed or mobile) and the layering sequence of the model (Figure 7a) were varied (Table 2). Shortening at 90° to the ramp represents the simplest possible scenario, whereas shortening at 45° to the ramp reflects oblique shortening during Cretaceous shortening. Moving the basal plate with the brittle-ductile materials on top represents thick-skinned shortening, whereas a fixed basal plate with movement only in the cover reflects thin-skinned shortening of the NCA. A varying layering with zero, one or two décollements tests the importance of a weak basal décollement and decoupling at the base or within the sedimentary cover. The modelling strategy evolved from a basic brittle model (A) using the simplest geometric, kinematic and lithological parameters possible. The same model set-up was then applied to brittle-ductile (B1-B3) models. The set-up featured a basal plate with dimensions of 40 x 22.5 cm, a thickness of 1 cm and a 60° dipping front ramp (Figure 7), representing the Jurassic normal fault. A wooden piston of 40 cm width was

placed behind (A, B1) or on top of (B2, B3) the basal plate, so that the plate was either mobile or fixed, exemplifying the effect of thick- versus thin-skinned tectonics. Shortening was applied at 90° to the front ramp with a convergence rate of 8 cm/h for the brittle model (A) and 1 cm/h for the brittle-ductile models (B1-B3), and a total shortening of 8 cm (Table 1). The convergence rates between brittle and brittle-ductile models vary, because the strength of the ductile material depends on e.g., the strain rate, whereas this is not the case for brittle materials. For the brittle model, only quartz sand was used as a modelling material. In the brittle-ductile models, a basal layer of silicone putty was placed on top of the basal plate (B1, B3) or over the basal plate, ramp and table top (B2). For technical reasons model B1 (and, for comparison, model B3) do not have silicone putty on the table top, because the material may stick beneath a mobile basal plate, creating modelling artefacts. Model set-ups with a fixed basal plate do not have this problem. A small amount of dishwashing soap was spread between the plate and the silicone putty, to prevent sticking of the ductile layer to the substrate. Model B2 featured an additional, upper ductile layer (Figure 7a), representing incompetent behaviour of the Carnian Raibl beds (Figures 3, 6).

For model C we added two plates to the north and south of the main basal plate, to simulate E-W striking strike-slip faults and expected sigmoidal fault arrangement in the subsurface of the Achental structure. This required the use of a piston with a width of 83 cm. The basal plates were fixed, representing a thin-skinned style of deformation. Shortening remained orthogonal to the ramp at 1 cm/h, but total shortening was increased to 11 cm to accommodate the larger size of the model set-up. The basal ductile layer of silicone putty was placed on top of the basal plates and on the table top, but was disconnected at the ramp (Figure 7) to create a clear velocity discontinuity as seen in model B3.

For models D1 and D2 the fixed main basal plate and two auxiliary plates were rotated 45°, to simulate Cretaceous oblique shortening at 45° to the ramp (Jurassic normal fault). We applied a total shortening of 11 cm, identical to model C. For both models, the basal ductile layer was placed on the basal plate and table top, while being disconnected at the ramp to create a velocity discontinuity as in model C (see e.g., Allemand & Brun, 1991; Tron & Brun, 1991). Model D2 featured an extra, upper ductile layer, simulating incompetent behaviour of the Raibl beds, similar to model B2.

**Table 2:** Summary of modelling runs and relevant parameters.

| Series | | Layering | Silicone | | Basal plate | Plate border | Angle (°) | Total shortening (cm) |
|--------|---|----------|# Layers | At ramp | | | | |
| A | | Brittle | 0 | --- | Mobile | Straigth | 90 | 8 |
| B | 1 | Brittle-ductile | 1 | Disconnected | Mobile | Straigth | 90 | 8 |
| | 2 | Brittle-ductile | 2 | Connected | Fixed | Straigth | 90 | 8 |
| | 3 | Brittle-ductile | 1 | Disconnected | Fixed | Straigth | 90 | 8 |
| C | | Brittle-ductile | 1 | Disconnected | Fixed | Sigmoidal | 90 | 11 |
| D | 1 | Brittle-ductile | 1 | Disconnected | Fixed | Sigmoidal | 45 | 11 |
| | 2 | Brittle-ductile | 2 | Disconnected | Fixed | Sigmoidal | 45 | 11 |

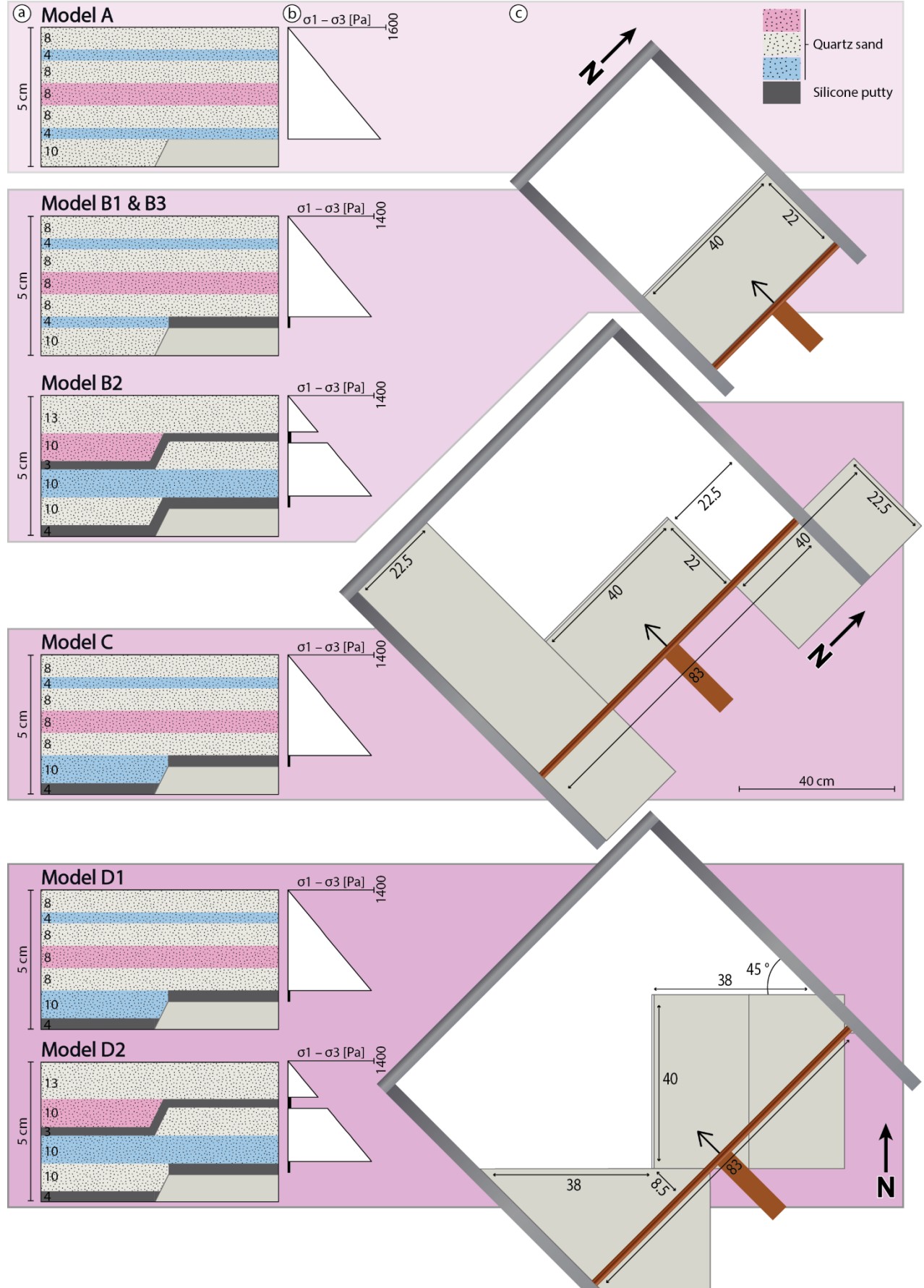

**Figure 7:** Modelling set-up for model series A-D. (a) Layering sequences showing differently coloured layers of quartz sand and silicone putty with a total thickness of 5 cm. (b) Strength envelops showing strength (in Pa) of different modelling materials. (c) Basal plate set-ups showing the basal plate size (in cm).

### 3.3 Monitoring and model analysis

After reaching the desired shortening, models were covered with black sand for protection, drenched with water until saturation and left to rest for at least 2 hours. Sections were then cut orthogonal to the strike of major structures. Because these sections only show the final result and structure of the models, particle image velocimetry (PIV) was used to analyse incremental displacements in the analogue models (e.g., Leever et al., 2011; van Gelder et al., 2017), using powdered coffee grains on the model surface as markers. For all models, photographs of the top surface of the experiment were taken in a fixed time interval of 30 minutes. Photographs were then rectified to correct for lens distortion and processed using the MATLAB® vR2022a application PIVlab v2.57 (Thielicke, 2014; Thielicke & Sonntag, 2021; Thielicke & Stamhuis, 2014). In the image pre-processing settings, we enabled Contrast-Limited Adaptive Histogram Equalization (CLAHE) to locally enhance the contrast. FFT window deformation was used as the PIV algorithm. As post-processing steps we applied a velocity vector validation using a local median filter and interpolated missing data points.

The "strainmap" package (Broerse, 2021) for MATLAB® (vR2022a) was used for calculating cumulative strain type maps for models C and D1 based on incremental displacements from the PIV analyses. For a review of the underlying algorithms and practical applications of this software, see Broerse et al. (2021) and Krstekanić et al. (2021; 2022). Resulting colour-coded maps show the cumulative strain type (extension, strike-slip movement or shortening) at four points during the model runs (see figures in Section 4).

### 3.4 Limitations and simplifications

Similar to other analogue (and numerical) modelling studies, a measure of simplification is necessary to create a modelling basis from the natural example. Therefore, one of the major limitations of the experiments in this study is the necessary simplification of the initial fault configuration. The Jurassic basin-bounding faults are presently not exposed at the surface and thus their geometry and exact position are not known. It is assumed that the major present-day structural elements are aligned to the margins of the Jurassic Achental basin (Ortner & Gruber, 2011; Sausgruber, 1994b; Töchterle, 2005). Arguments for the existence of such a basin are presented in Section 2.2. Following the geometrical considerations for this basin, we chose a simplified subsurface fault arrangement with one N-S and two E-W striking elements as the basis for the final series (C, D) of analogue models. Splay faults and additional basin-bounding faults are thus not considered here. The models were shortened in one single phase, simulating Cretaceous NW-directed shortening. However, the Achental structure was affected also by N-directed Paleogene and NE-directed Neogene shortening (Eisbacher & Brandner, 1995; Ortner & Gruber, 2011). These posterior phases of deformation were not taken into consideration for the analogue modelling, because they cannot explain the large offset along the Achental thrust in W–E sections (Ortner, 2003a; Ortner & Gruber, 2011) and are thus not expected to have created the main structural elements. The model set-up emphasizes the rheological contrast between a weak ductile basal décollement and a stronger upper brittle layer, whereas the characteristic, more complex heterogeneity of the NCA sedimentary cover was not included. Finally, the models ignore natural recovery processes such as erosion and sedimentation, which may influence the time–space evolution of structures. Eliminating limitations by increasing the complexity of models may increase the likeness between the analogue models and the natural example, but clouds the aim of analysing deformation localisation at pre-existing basin boundaries. Despite the simplifications described in this section, we are confident that our analoging results are meaningful for the problem under consideration.

### 4 Results

Model A is a purely brittle experiment with a maximum thickness of the sand layer of 5 cm. The piston pushed both the sand and the basal plate. In the initial stages of the experiment, an asymmetrical pop-up structure with a master back-thrust (1) and

an antithetic fore-thrust (1) formed (Figure 8). Further shortening was accommodated by transient fore-thrusts (2, 6-8), which developed at the base of the ramp and migrated upward through the wedge, along the master back-thrust..

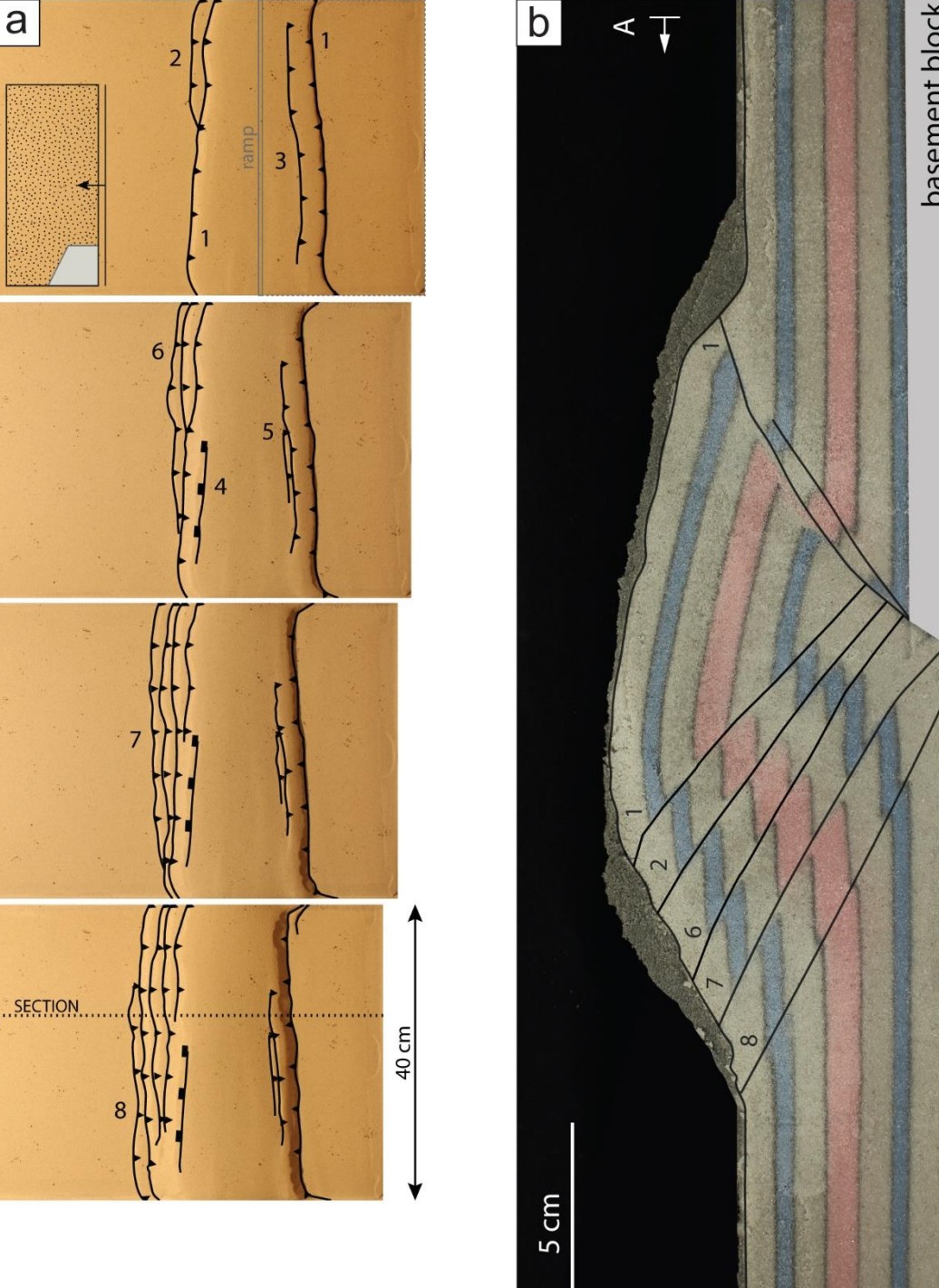

**Figure 8:** Modelling results of model A, showing (a) top views at 25, 50, 75 and 100 % of total shortening, and (b) side view at 100 % of total shortening. Location of section is marked in (a). Numbers show sequence of fault formation. The inset in the first top view photograph shows the modelling setup in cross-sectional view.

Model B1-B3 (Figures 9, 10) are brittle-ductile models with a basal décollement of silicone putty, designed to study coupling and decoupling processes at the basal plate and ramp. Model B1 initially developed an asymmetric pop-up structure with an antithetic fore-thrust (1) and a master back-thrust (2) similar to model A (Figures 9a, 10a). Although shortening was initially accommodated equally along the back-thrust and the fore-thrust, bulk shortening occurred along the master back-thrust. With further shortening, additional fore-thrusts (7, 9) form and former thrusts migrate upward along the master back-thrust. All

major thrusts originate from the velocity discontinuity at the ramp. Several secondary pop-up structures formed along splay faults (e.g., 3, 4, 6). The final resulting wedge is both wider and lower than in the brittle model A, and the internal wedge structure more complicated. Model B2 contained an additional, upper ductile layer. Furthermore, the basal ductile layer was draped over the entire model base and connected at the mobile ramp (Figure 7a). During the first 25% of total shortening, a central wedge developed over the ramp, formed by two main back-thrusts separated at the upper layer of silicone putty. At

~50% of total shortening hinterland deformation created an asymmetrical pop-up structure with a main back-thrust in the lower sand layers, and a symmetrical pop-up structure in the upper sand layer above the upper ductile horizon. Pockets filled with air that was trapped underneath the silicone during model construction represent modelling artefacts. Overall, the height of the resulting wedge was far less than in model B1 and structures were mainly hinterland-oriented and dominantly backwards thrusting (Figures 9b, 10b). Thrusts may have initially originated at the velocity discontinuity, but were laterally transported

onto the mobile basal plate as shortening progressed. The upper and lower sand layer were completely decoupled at the upper silicone layer. For model B3 we used an identical layering as for model B1, but the basal plate was fixed simulating thin-skinned deformation(Figure 7). In the first half of the experiment (0-50% total shortening), a flip-type pop-up structure (see Smit et al., 2003, their Figure 4) with two back-thrusts and conjugate fore-thrust (1) developed directly in front of the piston (Figures 9c, 10c). Movement occurred along the main back-thrust, switched to the fore-thrust and back to the secondary back-

thrust, as visible from lobes of silicone putty along the thrusts. At ~ 75% of total shortening, thrusting propagated into the foreland to form a strongly asymmetric pop-up structure with a master fore-thrust (5) and a minor back-thrust (6). The master thrust originated directly at the velocity discontinuity above the ramp. Although in the fixed plate scenario, as opposed to the mobile plate scenario, deformation did not localise directly above the ramp but rather required a wedge in the hinterland, the wedge geometry is much more asymmetric with most of the shortening occurring along a low-angle master fore-thrust that is

antithetic to the original normal fault geometry.

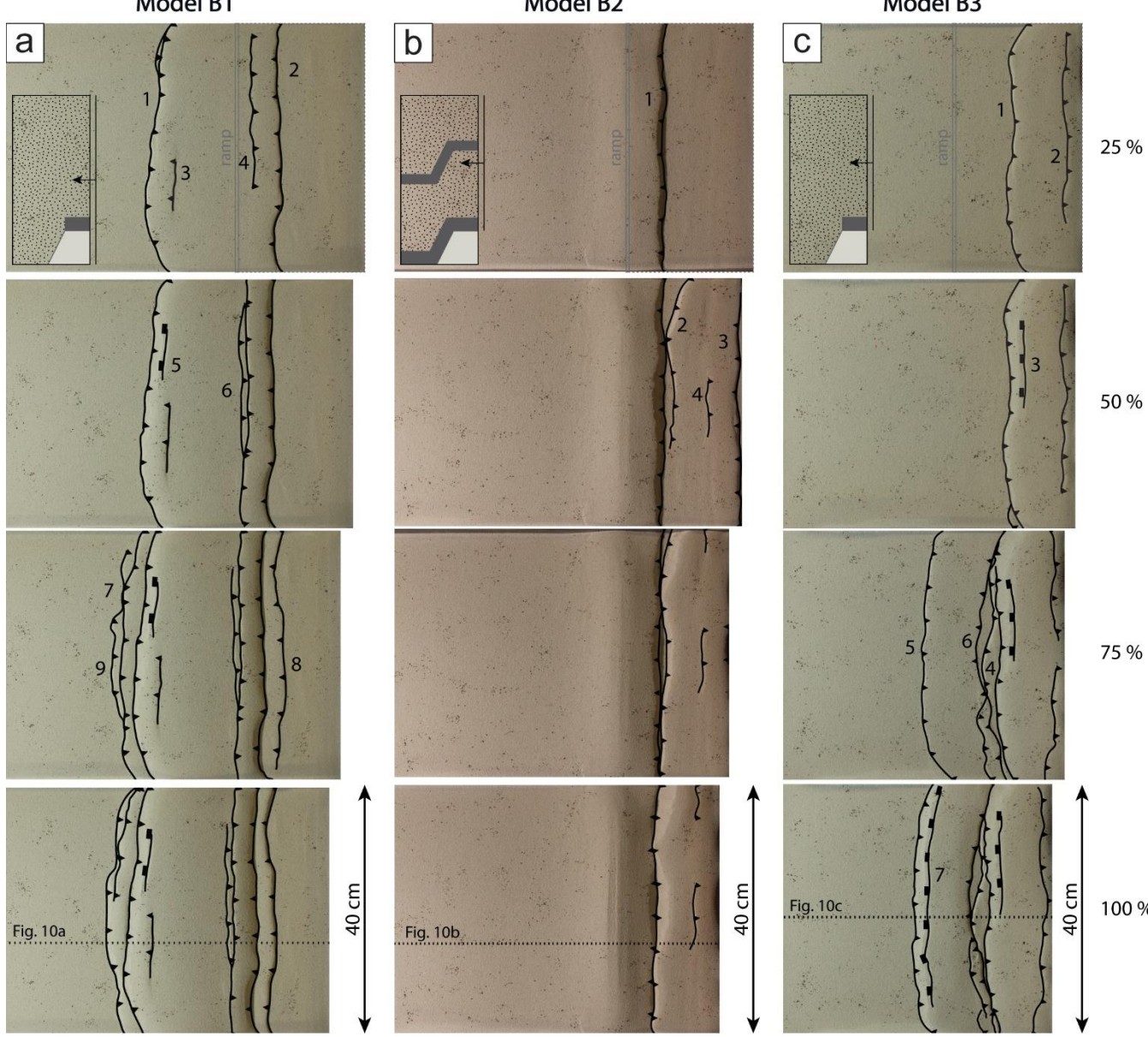

**Figure 9:** Top views of models (a) B1, (b) B2, and (c) B3 at 25, 50, 75 and 100 % of total shortening. Locations of sections are indicated (dotted lines). Numbers show sequence of fault formation. The insets in the first top view photographs show the modelling setups in cross-sectional view.

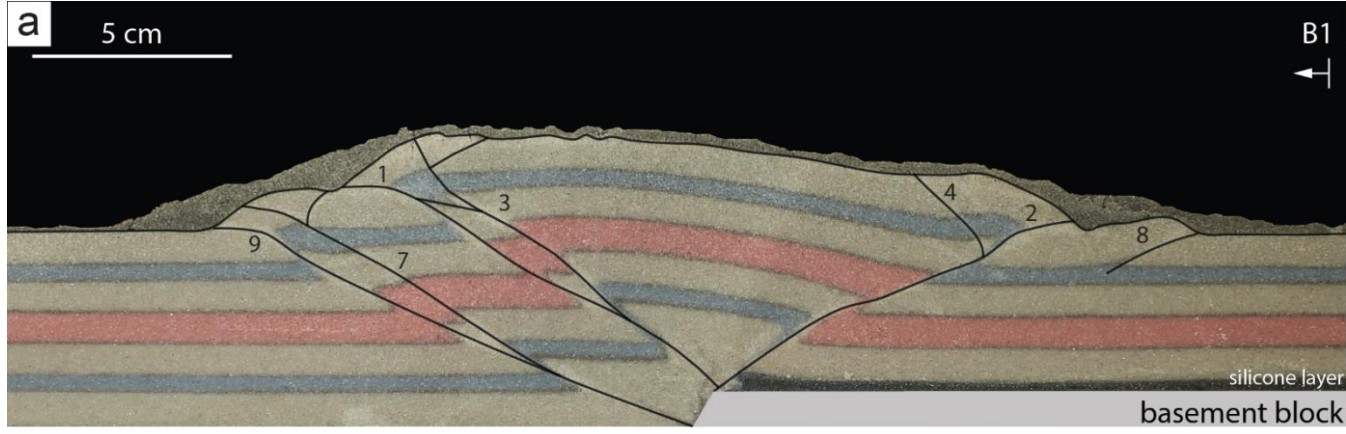

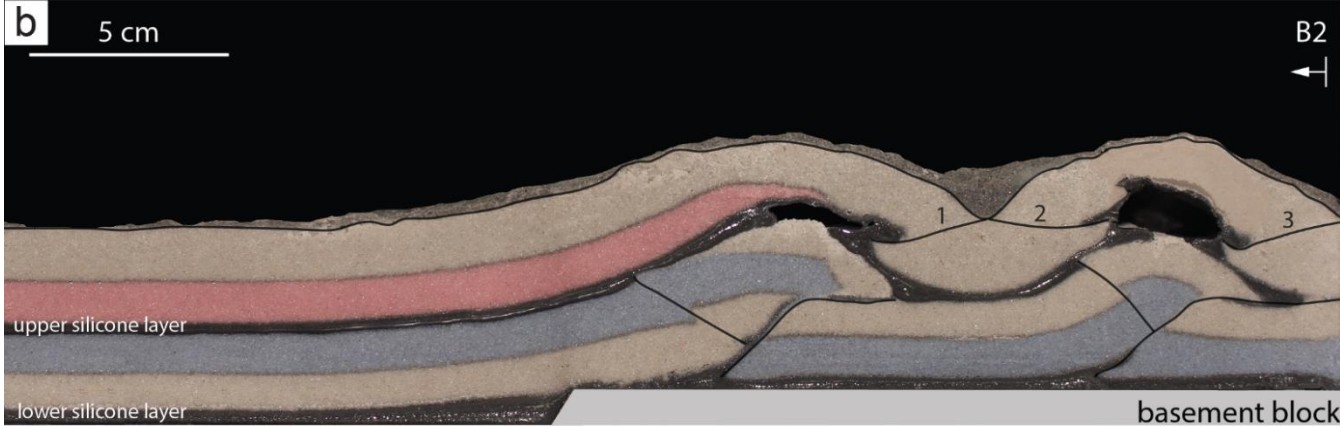

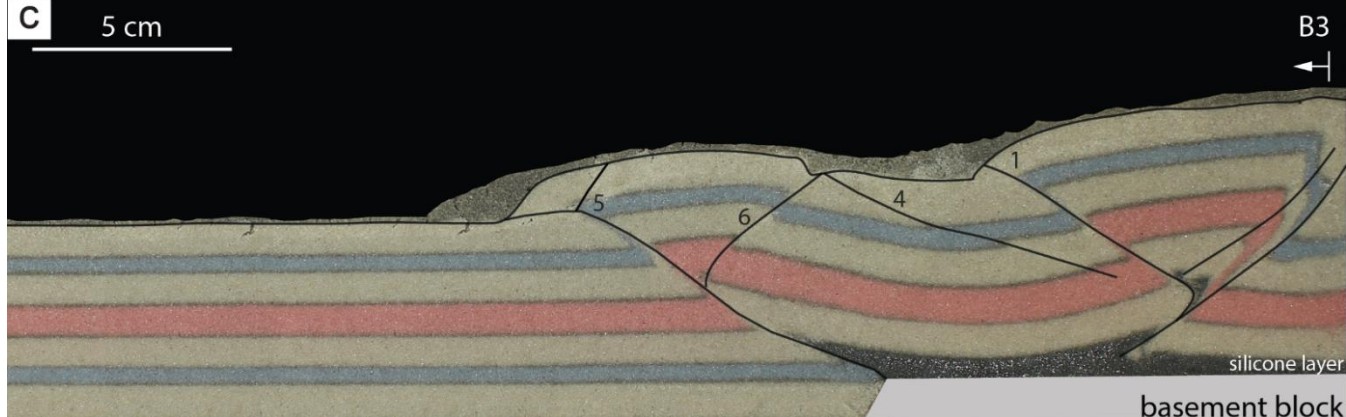

**Figure 10:** Side views of models (a) B1, (b) B2, and (c) B3 at 100 % of total shortening. Locations of sections are marked in Figure 9. Numbers show sequence of fault formation.

For model C, we used a basal plate geometry with three basal plates, aligned in an s-form (Figure 7c). We placed silicone putty on both the basal plates and the table top, disconnecting it at the ramp to form a similar mechanical scenario as in model B3. Initially, a slightly asymmetrical pop-up structure formed close to the piston, with a back-thrust that migrated slightly upward along the main fore-thrust (2) (Figure 11a-b). Additional fore-thrusts (1, 3) developed especially in the northern area of the model, where the basal plate was missing. After ~ 55% of total shortening, deformation propagated into the foreland, where a second asymmetric pop-up structure formed above and parallel to the ramp in the subsurface. The master fore-thrust (5) accommodated most of the shortening, whereas several minor back-thrusts migrated upward along the master thrust. The sand layer forms a ramp anticline over the master thrust, which approaches lower angles toward the surface. Close to reaching the total shortening, deformation propagated in the southern area of the model, where the basal plate reaches into the foreland, and third pop-up structure was formed there (9), but was not able to develop further with this amount of shortening. Overall, the

wedge was relatively low compared to the brittle model A and the brittle-ductile model B1. The deformation front at total shortening approximately shows the contours of the basal plate in the centre and southern parts of the model (Figure 11a).

PIV analysis of (surficial) shortening shows that deformation occurs in the tectonically active region west of the deformation front, with strain localisation at the foremost thrust (2, 5, 9/10). Strain directions rotate outward on both the northern and southern side of the wedge. The dominant type of strain at the active fore-thrusts is shortening (red colours in Figure 11c). Extensional strain on top of the wedge (blue colours in Figure 11c) is mainly caused by gravitational collapse of parts of the wedge. We attribute local strike-slip strain (indicated by yellow to turquoise colors in Figure 11c) to erroneous vectors

generated by the PIV analysis in areas where individual sand grains could not be tracked optically.

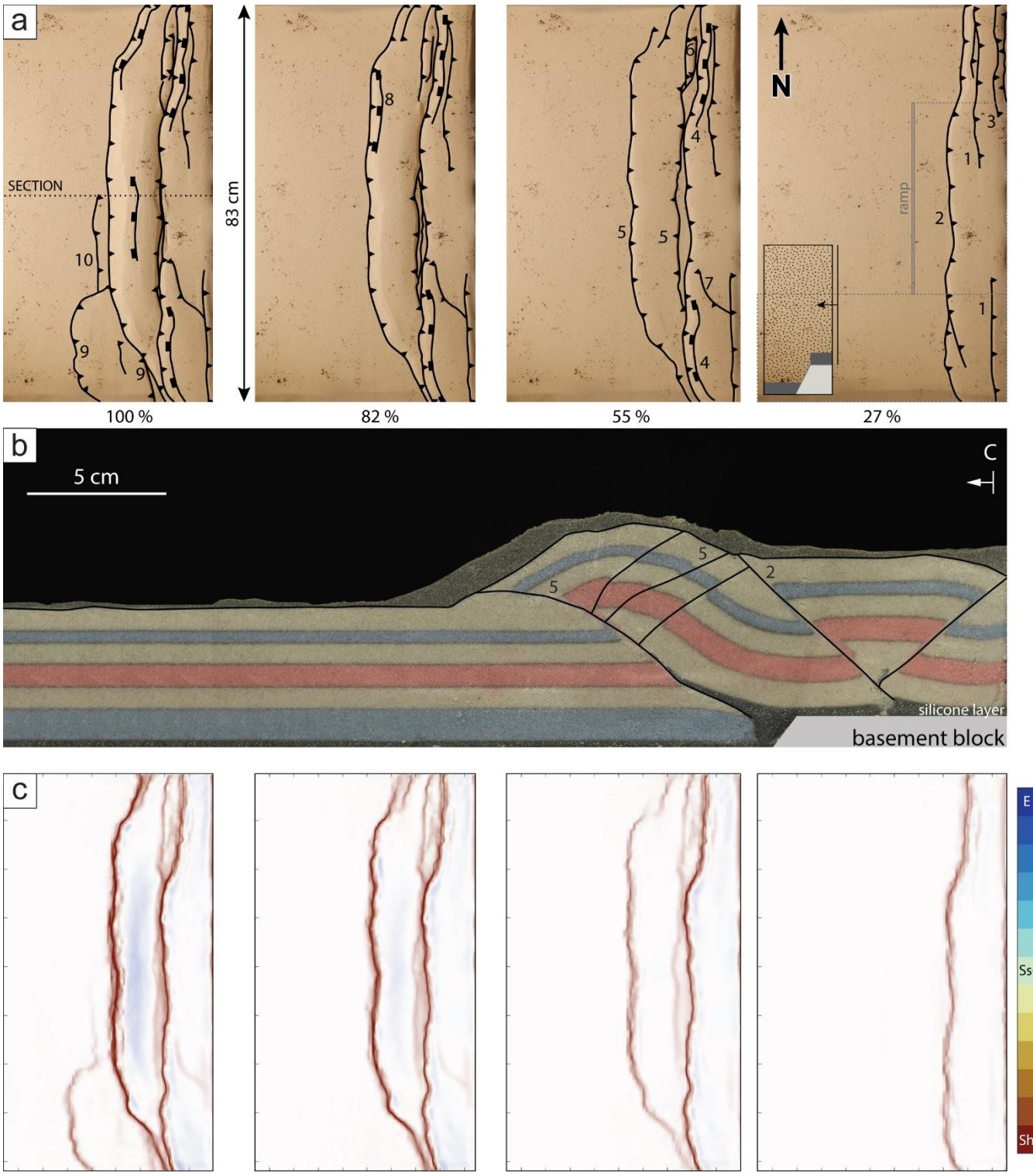

**Figure 11:** Modelling results of model C, showing (a) top views at 27, 55, 82 and 100 % of total shortening, and (b) side view at 100 % of total shortening. Numbers show sequence of fault formation. (c) Analysis of cumulative strain showing strain types of model C. Areas of extension (E), strike-slip movement (Ss) and shortening (Sh). The inset in the first top view photograph shows the modelling setup in cross-sectional view.

For model D1 and D2 shortening was directed at an angle of 45° to the basal plate and ramp (Figure 7c). Model C and D1 used an identical layering sequence. In the initial stages of the experiment (< 55% total shortening), a pop-up structure developed over the entire length of the piston (Figure 12a). This structure is asymmetric with a main fore-thrust in the south. In the central area of the model, where the basal plate set-up creates a 90° angle, the structure is of a flip-type (see Smit et al., 2003, their Figure 4), where shortening was initially accommodated along a fore-thrust, before switching to a back-thrust. From approximately 27% of total shortening on, the wedge widened along the main fore-thrust (1) in the south, where the basal plate extends into the foreland, whereas in the centre and north of the model, shortening was accommodated by a series of fore-thrusts (3-5) that migrated along an associated back-thrust. From ~ 82% of total shortening, the wedge propagated on the southern basal plate (7), and then on top of the northern plate (8). The surface expression of the propagating fore-thrusts runs parallel to the plate structure in the subsurface. The thrusts originate directly at the velocity discontinuity at the ramp (Figure 13a-b) and form low-angle (~30°) master thrusts in strongly asymmetric pop-up structures, similar to model B3. At the 90° corner between basal plates a steepening of the back thrust and migration of irregularly spaced fore-thrusts within the pop-up structure accommodate shortening.

PIV analysis of incremental displacements shows that shortening localises at the foremost thrust in model D1 (1, 7). The strain type patterns are similar to model C, with shortening as the dominant strain type, and the dominant strain direction perpendicular to the thrust trace. The orientation of the basal plates in the subsurface evidently controls the orientation of the thrusts, with a local rotation of the strain field.

Model D2 used the same basal plate set-up as model D1, with a layering sequence using two ductile layers, similar to model B2 (Figure 7). Similar to model B2, the upper and lower part of the model are decoupled at the upper ductile layer. Propagation into the foreland occurred at~55% of total shortening. In the lower part of the model, pop-up structures with a master fore-thrust localised at the ramp (Figures 12c, 13c). These structures form a sigmoidal shape that straddles the basal plate structure in the subsurface. The decoupling at the upper ductile layer caused that the surface structures differ from the lower part of the model and the wedge reached far less height than in experiments with a single ductile layer.

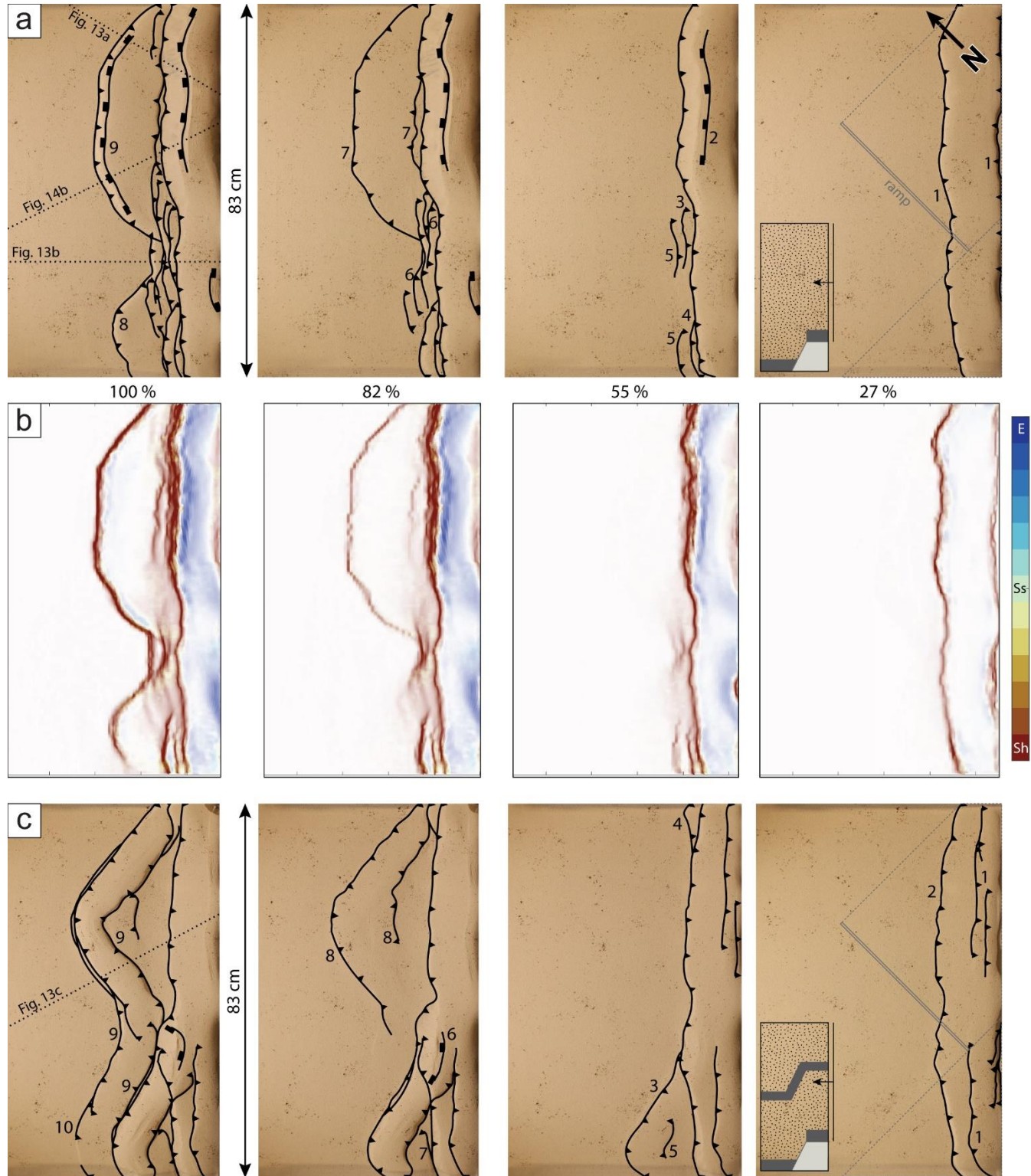

**Figure 12:** Top views of models (a) D1, and (c) D2 at 27, 55, 82 and 100 % of total shortening. Numbers show sequence of fault formation. The insets in the first top view photographs show the modelling setups in cross-sectional view. PIV strain type analysis of model (b) D1, showing areas of extension (E), strike-slip movement (Ss) and shortening (Sh).


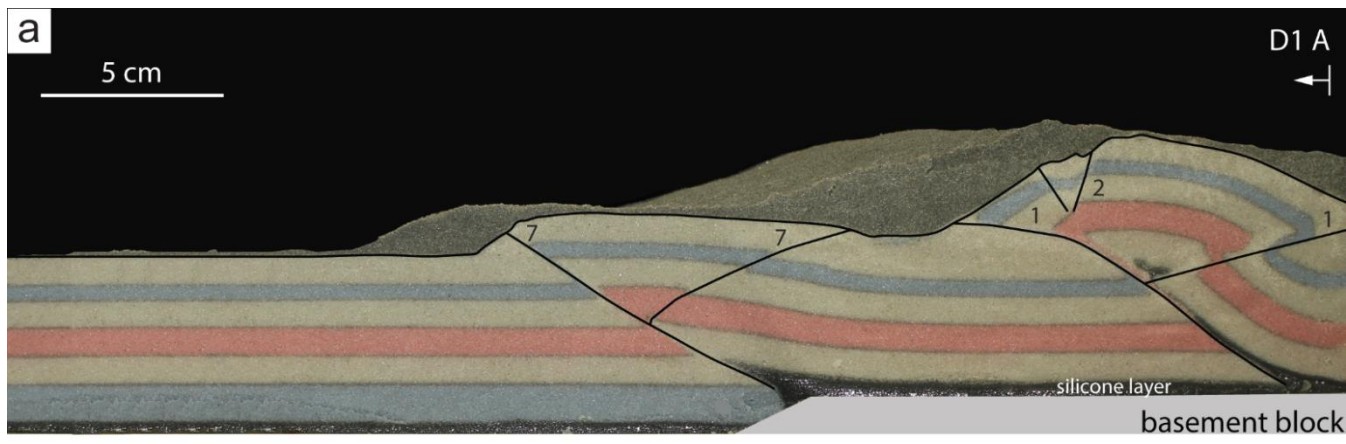

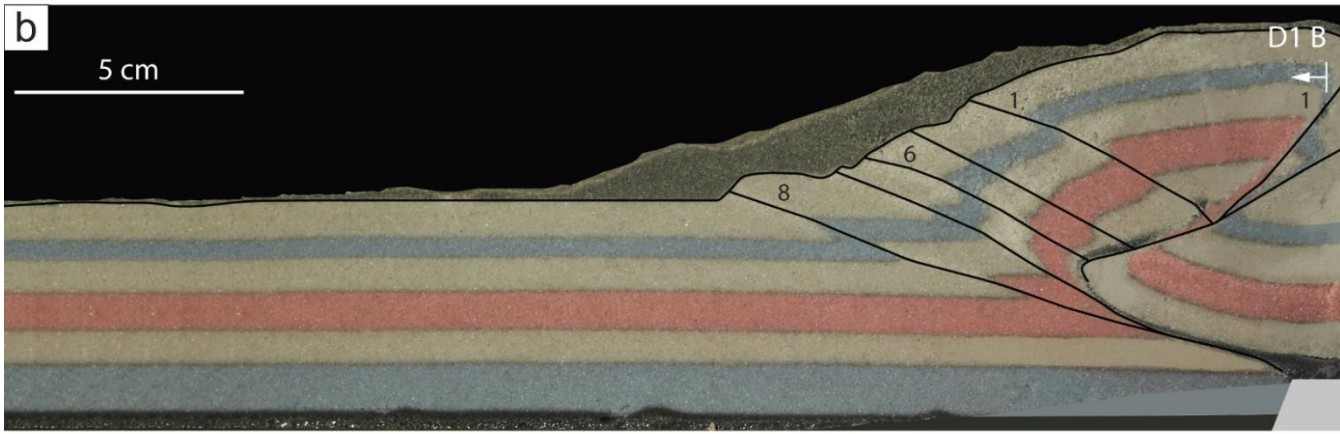

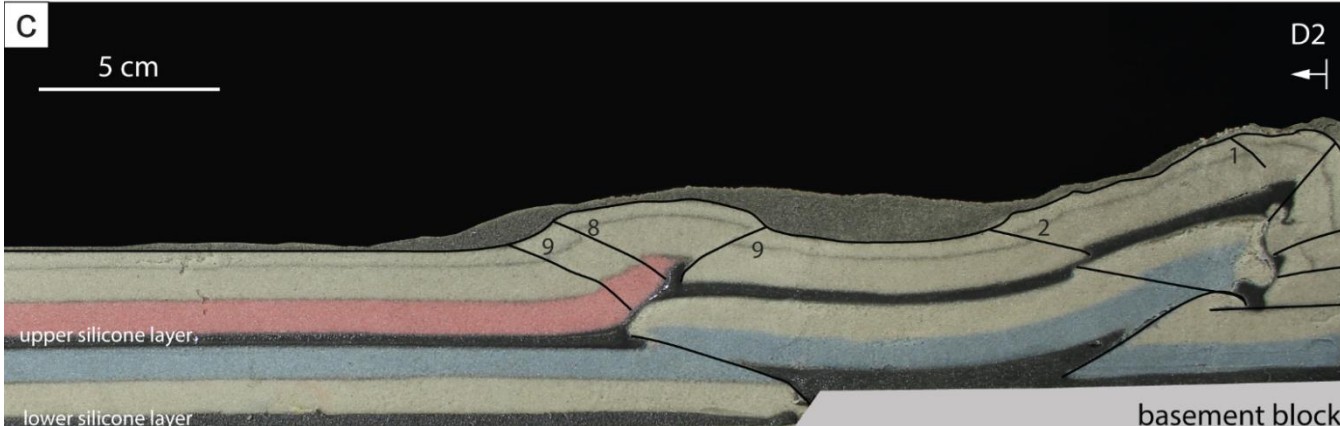

**Figure 13:** Side views of models (a, b) D1, and (c) D2 at 100 % of total shortening. Numbers show sequence of fault formation.

## 5 Discussion

### 5.1 Analogue modelling

Inversion of basins, whether orthogonal or oblique to the former basin margins, has been subject of many analogue modelling studies (e.g., Bonini et al., 2012; Brun & Nalpas, 1996; Del Ventisette et al., 2006; Deng et al., 2020; Molnar & Buiter, 2023; Sieberer et al., 2023; Yagupsky et al., 2008; Zwaan et al., 2022). The high variance of geological structures leads to a myriad of deformation styles characterizing inverted and reactivated fault systems (Bonini et al., 2012; their Figure 3), especially for inversional set-ups involving a viscous décollement (Brun & Nalpas, 1996). The modelling strategy (see Section 3.1) followed in this study aimed to test factors that may affect the structural grain of obliquely inverted extensional basins, including the influence of 1) a weak basal décollement, and 2) pre-existing structures attributed to an extensional basin. The increasing complexity of the models is a result of consecutively implementing parameters that were found to be of relevance to the natural

example, the Achental structure. Apart from its sigmoidal hanging wall shape, one of the complexities of the Achental structure is that its dominant fore-thrust is antithetic to the Jurassic normal fault in the subsurface (Figure 6). Most of the structures associated with basin inversion are synthetic to existing normal faults (e.g., Héja et al., 2022: their Figure 4 and 11; Bonini et al., 2012: their Figure 3), although the formation of an antithetic low-angle thrust has been described (Laubscher, 1986;
Tavarnelli, 1996) and simulated in the context of thin-skinned deformation of the Jura fold and thrust belt (Caër et al., 2018).

### 5.1.1 Influence of mechanical stratigraphy

In our study, the influence of a weak basal décollement has been tested using two models (A and B1). These have an identical mechanical set-up, using a rigid ramp, which represents a pre-existing normal-fault controlled mechanical heterogeneity in the model and is pushed into a brittle (A) or brittle-ductile (B1) cover. The brittle-ductile model (B1) shows a narrower cross-
sectional taper, a markedly lower angle of thrusting, wider thrust spacing and faster propagation compared to the brittle model (A). Decoupled thrust wedges are known to show these effects in nature (Jaumé & Lillie, 1988) and analogue models, attributed to a lower basal friction compared to frictional Coulomb wedges (Cotton & Koyi, 2000; Davis & Engelder, 1985; Mulugeta, 1988; Smit et al., 2003; Smit, 2005). Thus, our brittle-ductile model B1 is able to better represent the low-angle character of the natural example than model A.

Models of oblique basin inversion show that the use of a ductile layer results in decoupling of the sedimentary cover and a strong localisation of deformation at existing extensional structures (e.g., Brun & Nalpas, 1996; Del Ventisette et al., 2006). Similarly, the use of an upper (second) ductile layer (models B2, D2) leads to a decoupling between the upper and lower sediment cover (e.g., Del Ventisette et al., 2006; Fan et al., 2020). As a result, structures within the lower brittle layer are terminated at the upper ductile layer. Low-angle hinterland-dipping thrusts emerging from the velocity discontinuity are
therefore not visible in top view (see e.g., model D2, Figure 13). Decoupling thus prevents the formation of large-scale structures across the entire sedimentary cover.

### 5.1.2 Favourable kinematic conditions for basin inversion

The build-up of a certain height of the initial wedge at the backstop is a prerequisite for fault propagation, consistent with the critical taper theory of Davis et al. (1983) (see also Graveleau et al., 2012, and references therein). In all experiments, a wedge
of 5.7–7.6 cm high was formed in front of the piston and at least 6 cm of shortening was needed for the thrusts to propagate into the foreland. The movement of the basal plate (i.e., fixed or mobile, simulating thin- or thick-skinned deformation) then greatly influences the structure and location of the resulting wedge in brittle-ductile models (compare e.g., model B1 and B3). By using a mobile plate (model B1), this rigid block acts as a buttress, localising early deformation at the ramp and restraining further movement of thrust sheets (Bailey et al., 2002; Gomes et al., 2010; Héja et al., 2022). Further shortening was
accommodated along a main back-thrust with multiple fore-thrusts, showing strong similarities to existing analogue experiments (e.g., Bonini et al., 2000; Persson & Sokoutis, 2002).. The use of a fixed plate (model B3) leads to initial strain localisation at the backstop (see also Caër et al., 2018; Deng et al., 2020; Yagupsky et al., 2008). The mechanical contrast between the rigid ramp, quartz sand and silicone putty forms a velocity discontinuity (Allemand & Brun, 1991; Tron & Brun, 1991), which is crucial for secondary strain localisation. The step at the ramp, where silicone putty and quartz sand meet, thus
works as a "fault generator" or a "nucleation site for thrust faulting" (Yagupsky et al., 2008, p. 852), not dissimilar to the "thrust mill" of Laubscher (1986), which is also demonstrated in models that use silicone putty or glass microbeads as a weak layer (Caër et al., 2018; Yagupsky et al., 2008). At the same time, structures localising at the ramp accommodate a large amount of shortening, preventing deformation from propagating further (Caër et al., 2018; Laubscher, 1986; Tavarnelli, 1996). A low-angle fore-thrust nucleating at the velocity discontinuity thus may accommodate a great amount of shortening.


### 5.1.3 Influence of structural inheritance

Propagation of deformation and strain localisation in brittle-ductile models led to the formation of structures oriented parallel to the basal plate boundaries in the subsurface. Other studies of oblique shortening of pre-existing grabens or steps show similar results, where newly formed reverse and thrust faults more or less outline the structure in the subsurface (e.g., Caër et al., 2018; Deng et al., 2020; Molnar & Buiter, 2023; Yagupsky et al., 2008). Whereas thrusts in models of Caër et al. (2018) appear to consistently dip towards the backstop., Our PIV analyses show that the dominant strain direction is approximately perpendicular to the thrust trace, which can be explained by strain partitioning mechanisms. At the same time, the localisation of structures at steps and the geometry of the resulting low-angle fore-thrust with a hanging wall anticline (e.g., model D1, Figure 13a–b) is similar to the models of Caër et al. (2018; their Figure 4.11 b–e) and it is evident that in models with oblique shortening (model D1, D2) thrust faults outline the basal plate structure.

The obliquity angle between the basin axis and subsequent shortening greatly influences structures formed by basin inversion (Brun & Nalpas, 1996; Del Ventisette et al., 2006; Deng et al., 2020; Yagupsky et al., 2008). In brittle experiments, low obliquity angles are associated with an increasing angle of the reverse faults (Brun & Nalpas, 1996) and dominant fore-thrusts, whereas a higher obliquity angle preferably creates symmetrical pop-up structures (Deng et al., 2020). Our brittle-ductile models were able to create a dominant fore-thrust that localizes at the ramp regardless of the obliquity angle (90° for models B–C, 45° for models D). However, fault localisation at subsurface steps was most distinctive in models with a lower obliquity angle (45°, models D). Our large-scale models (C and D) furthermore agree with those of Yagupsky et al. (2008) and Molnar and Buiter (2023), where the graben segment closest to the deformation front is reactivated first.

The analogue models (in particular models B3, C, D1; Figures 9c, 10b and 12a) show that low-angle hinterland-dipping thrusts can originate at a pre-existing high-angle normal fault with an opposite dip (see also Figure 6), as predicted (Laubscher, 1986; Tavarnelli, 1996). The pre-existing subsurface fault network is able to control the geometry of inversion structures. The models exemplify that through strain localisation at subsurface steps, the sigmoidal geometry of the Achental thrust hanging wall (see model series D, Figure 12) can be created within a single deformation phase of oblique shortening.

## 5.2 Comparison with the natural example

In analogue models we aim to create a scaled and simplified version of the natural example, the Achental structure. We show that it is possible to form a sigmoidal structure, characterized by a low-angle main fore-thrust, hanging wall anticlines and a localisation at a pre-existing step, within a single phase of shortening (Figure 14). Model parameters that were found to be applicable to the Achental structure are 1) a weak basal décollement, 2) thin-skinned deformation, and 3) a clear velocity discontinuity acting as a "fault generator". However, laboratory experiments cannot fully encompass the complexities of natural geological structures. In this section, we compare features of the analogue models and the Achental structure.

### 5.2.1 Mechanical stratigraphy

Rheological heterogeneity of sedimentary successions are important parameters for both analogue models and natural processes. A brittle-ductile succession is able to re-create large-scale structures in analogue models. However, small-scale faulting and folding within the sedimentary cover of the NCA is cannot be reproduced by the analogue models (see also Figure 14b). Comparison between brittle and brittle-ductile models shows that a very weak décollement is imperative for the low-angle Achental thrust to form at a pre-existing basement step. Whether or not this décollement is present in the hanging wall of the original normal fault, does not influence strain localisation in the models, although it might impact fault propagation with further shortening (Caër et al., 2018). An offset of the décollement, as opposed to a monocline across the ramp (model B2), increases the effect of the velocity discontinuity, but is not strictly necessary for localisation to occur. The Haselgebirge-Reichenhall succession at the base of the Karwendel thrust sheet forms the main décollement of the NCA (Eisbacher &

Brandner, 1995). An intermediate succession of Carnian shales, evaporites and carbonates (Raibl Fm) may also be represented by a ductile layer, decoupling the lower and upper carbonate platform. Kilian et al. (2021) argue that in this part of the NCA, the Carnian units mostly consist of carbonates (e.g., Brandner & Poleschinski, 1986; Jerz, 1966), which we modelled as competent units. Our analogue models with two detachment horizons show a complete decoupling between the upper and lower brittle section. Although in the Achental structure, small-scale folding dies out near the Hauptdolomit-Plattenkalk transition, the overall geometry of the Unnutz anticline is preserved through the entire outcropping sedimentary succession. Therefore, we dismiss the hypothesis that the Raibl Fm behaves as a significant decoupling horizon.

Carbonate units within the Achental structure are represented by quartz sand in analogue models, simulating brittle behaviour. This modelling material is not able to simulate folding of the Guffert-Unnutz-Montscheinspitze anticline. Resulting from the chosen modelling materials and scaling, important features of the natural example, e.g., the large overturned panels seen in the cross-sections (Figures 5 and 14), cannot be explained by the analogue models. However, the thinning of strata in the hanging wall, close to the fault, could be reproduced (Figure 14b). Ortner (2003a) proposes a progressive rollover-fault-propagation model (Storti & Salvini, 1996) for the Unnutz anticline. This type of folding leads to strongly overturned or recumbent anticlines in fold-and-thrust belts (Storti & Salvini, 1996) and involves a fault-propagation fold with strong, layer-parallel shear (flexural slip). Although for fault-propagation folds "important field evidence […] is the observation that some faults, particularly thrust faults, die out in the cores of folds." (Suppe, 1985, p. 350), this is not necessarily the case for progressive rollover-fault-propagation folds, because the overturned limb of the hanging wall anticline has been completely detached from the footwall (Storti & Salvini, 1996: Fig. 2). Flexural slip accommodating deformation within the Guffert-Unnutz-Montscheinspitze anticline is seen from layer-parallel and very low-angle fault planes within Hauptdolomit, and support the progressive rollover-fault-propagation model. Fault-propagation folding as a mechanism is seen in the La Roche d'Or anticline, Swiss Alps (Caër et al., 2018), which formed along a thrust fault originating at a basement. Similar to the Reichenhall-Haselgebirge succession in the NCA, evaporitic sediments form a décollement between the basement and the sediment cover. Martin and Mercier (1996) provide a natural example of the western Jura, showing the development of a low-angle ramp above pre-existing structures of the Bresse graben. While the progressive rollover-fault-propagation model produces overturned panels, it does not explain e.g., the hinge collapse interpreted for the Unnutz anticline (Figure 5a). Although thinning of the overturned limb is expected and can be seen in the analogue models as well (Figure 14b), the complete elimination of stratigraphy in the overturned flank of the anticline cannot be explained by this type of folding.

In salt-detached fold-and-thrust belts the structural style is among others dependent on the amount of salt available (Hudec & Jackson, 2007; Lacombe et al., 2019). Within salt-controlled belts, anticlines are mostly (faulted) detachment folds, which can be mistaken as fault-propagation folds due to their superficial resemblance (Mitra, 2002). Detachment folding (e.g., Epard & Groshong, 1995; Homza & Wallace, 1995; Josep Poblet, 1996) and subsequent truncation and displacement of these folds (e.g., Jamison, 1987; Morley, 1994; Suppe & Medwedeff, 1990) produces asymmetric anticlines with steep to overturned forelimbs (Wallace & Homza, 2004). In the case of salt-related structures (e.g., minibasins), shortening will initially concentrate on these structures (Duffy et al., 2018; Snidero et al., 2019). Reactivation of salt welds as thrusts and rotation of flaps may produce overturned panels with incomplete stratigraphy (Duffy et al., 2018; Granado et al., 2019; Granado et al., 2021; Rowan & Vendeville, 2006). The exact type of deformation strongly depends on the original thickness of the Haselgebirge-Reichenhall succession. In the case of the Achental structure is it unknown how much salt was present originally. If a substantial amount was available, it might have controlled the geometry of overlying younger deposits. However, no major growth wedges or angular unconformities have been observed in the area up to now.

## 5.2.2 Kinematic parameters and structural inheritance

In analogue models, the use of a fixed basal plate translates to a thin-skinned tectonic style, in which the sedimentary cover is decoupled from the basement by a décollement (Rodgers, 1949). Such thin-skinned structural style is well known from the NCA (e.g., Auer & Eisbacher, 2003; Eisbacher et al., 1990). For localisation and propagation of deformation, the presence of a step in the subsurface and topography in the hinterland is important as well. The advancing front of the NCA fold-and-thrust belt and the uplift resulting from movement along e.g., the Eben thrust (Eisbacher & Brandner, 1996) (Figure 1b) may have provided this topography. The Jurassic basin architecture in the subsurface is thought to have provided steps where deformation could localise (Figure 14a). Facies differentiations of Jurassic sediments, a maximum thickness of Upper Jurassic sediments in the overstep area between the Karwendel and Thiersee synclines (Figure 1c) (Nagel et al., 1976; K.-I. Schütz, 1979), and documented Cretaceous transport directions oblique to the synclines (Ortner & Gruber, 2011; Sausgruber, 1994a, 1994b) underline the presence of such a basin (see Section 2.2). We assume that, corresponding to the geometry of the Achental structure, at least one N–S striking, west-dipping normal fault and two E–W trending strike-slip faults must have existed (Figure 14a). Such a basin geometry is similar to Jurassic fault systems in more western parts of the Alpine orogen (Eberli, 1985, 1987; Weissert & Bernoulli, 1985) and has been proposed for the Achensee region (Channell et al., 1990; Eisbacher & Brandner, 1995, 1996).

The models were subjected to a single phase of oblique shortening, simulating NW-directed Cretaceous convergence related to Alpine orogeny. Field data from the Achensee area (Beer, 2003; Sausgruber, 1994b) support NW-directed shortening. Ortner and Gruber (2011) argue that the offset in the central Achental thrust is larger than in its northern and southern thrust segments, and therefore the direction of shortening must have been < 270°. Because the orientation of the folds (e.g., the Thiersee syncline; Töchterle, 2005) and faults within the Achental structure is oblique to Cretaceous shortening (e.g., Eisbacher & Brandner, 1996), many studies have offered alternative hypotheses involving multiple deformation phases and directions (e.g., Ampferer, 1921; Auer, 2001; Channell et al., 1990; Channell et al., 1992; Fuchs, 1944; Ortner, 2003a; Spengler, 1953). Although the formation of the Guffert-Unnutz-Montscheinspitze anticline and the Achental thrust is ascribed to pre-Gosau, deformation (Ortner & Gruber, 2011; Töchterle, 2005), this is no proof that the entire sigmoidal hanging wall formed during a NW direction of shortening. However, in our models we re-create the geometry of the Achental structure (including a low-angle thrust antithetic to the assumed subsurface step, and a a surface expression of the hanging wall outlining the subsurface geometry) by applying shortening at 45° to the N–S trending Jurassic fault, demonstrating that a single uniform phase of oblique shortening is sufficient to create a sigmoidal shape of the hanging wall that outlines the subsurface basin geometry of the Achental structure, as depicted in Figure 14. Our results thus agree with the hypothesis that the characteristic shape of the Achental structure formed due to forced folding at the borders of a Jurassic basin (Eisbacher & Brandner, 1995, 1996; Ortner & Gruber, 2011).

## 5.3 Limitations

The NCA are a salt-influenced fold-and-thrust belt with at least one (basal) décollement consisting of a pre-rift evaporitic succession. We chose to model this décollement as a relatively thin layer of iron powder-dosed silicone putty at the base of a thick, brittle sedimentary cover. However, geological settings often do not show ideal ("layer-cake") stratigraphic geometries. In salt-influenced regions, diapirs may create e.g., minibasins or flaps (e.g., Rowan et al., 2016). Using non-layer-cake geometries in physical experiments produces structures that are much different from classical thin-skinned fold-and-thrust belts (e.g., Rowan & Vendeville, 2006) and thus the classical layer-cake approach may lead to misinterpretation (Lacombe et al., 2019). In the central and eastern NCA the role of salt tectonics has been addressed by an increasing number of (recent) studies (e.g., Fernández et al., 2021; Fernández et al., 2022; Granado et al., 2019; Granado et al., 2021; Santolaria et al., 2022; Strauss et al., 2021). Resulting structural styles are highly complex and include km-scale overturned panels and missing

stratigraphy (Granado et al., 2019). The amount of salt available in salt-influenced fold-and-thrust belts controls the type of structures that can form during inversion (Hudec & Jackson, 2007; Lacombe et al., 2019). Triassic growth strata caused by salt tectonics have been reported for the central and eastern NCA, and locally in the western NCA (Fernández et al., 2021; Fernández et al., 2022; Granado et al., 2019; Granado et al., 2021; Kilian et al., 2021; Kilian & Ortner, 2019; Ortner & Kilian, 2022; Santolaria et al., 2022; Strauss et al., 2021), but there are no reliable estimates on the thickness of the Haselgebirge-Reichenhall succession underneath the Achental structure. The main goal of our modelling approach was to gain insight in conditions controlling strain localization and associated folding and thrusting at pre-existing basin boundaries in an oblique convergent setting. We did not aim to include the complexity of salt tectonics in our models, therefore a silicone putty has been mixed that does not favour diapir formation. Introducing Triassic salt dynamics to our models would then create e.g., flaps and minibasins. Hypothetically, if the overturned panels originally were flaps onlapping salt diapirs, localisation of folding and thrusting would have been easier, as diapirs would localize thrusts. In such a case, the Jurassic basin geometry would have an even more profound effect on structures at the original basin margins.

Another limitation of the analogue models compared to the natural example is that we did not consider Paleogene reactivation of Cretaceous structures. Paleogene folding however, is common throughout the Achental structure and Achensee area. E.g., the Hofjoch, Blaubergalm and Seekar folds (Figure 4) (Brandner & Gruber, 2011; Eisbacher & Brandner, 1995; Sausgruber, 1994b) show an orientation corresponding to post-Cretaceous shortening. The Seekar folds are offset by the N–S striking Seekar fault, which shows dextral strike-slip movement and a P-axis that corresponds to NNE- to NE-directed Paleogene shortening. Paleogene reactivation of fault planes also occurs at the west side of the Achensee (Sausgruber, 1994b). Although this post-Cretaceous reactivation may have imprinted on existing structures, we propose that the rough outlines of the Achental structure were established after Cretaceous orogeny (Figure 14a).

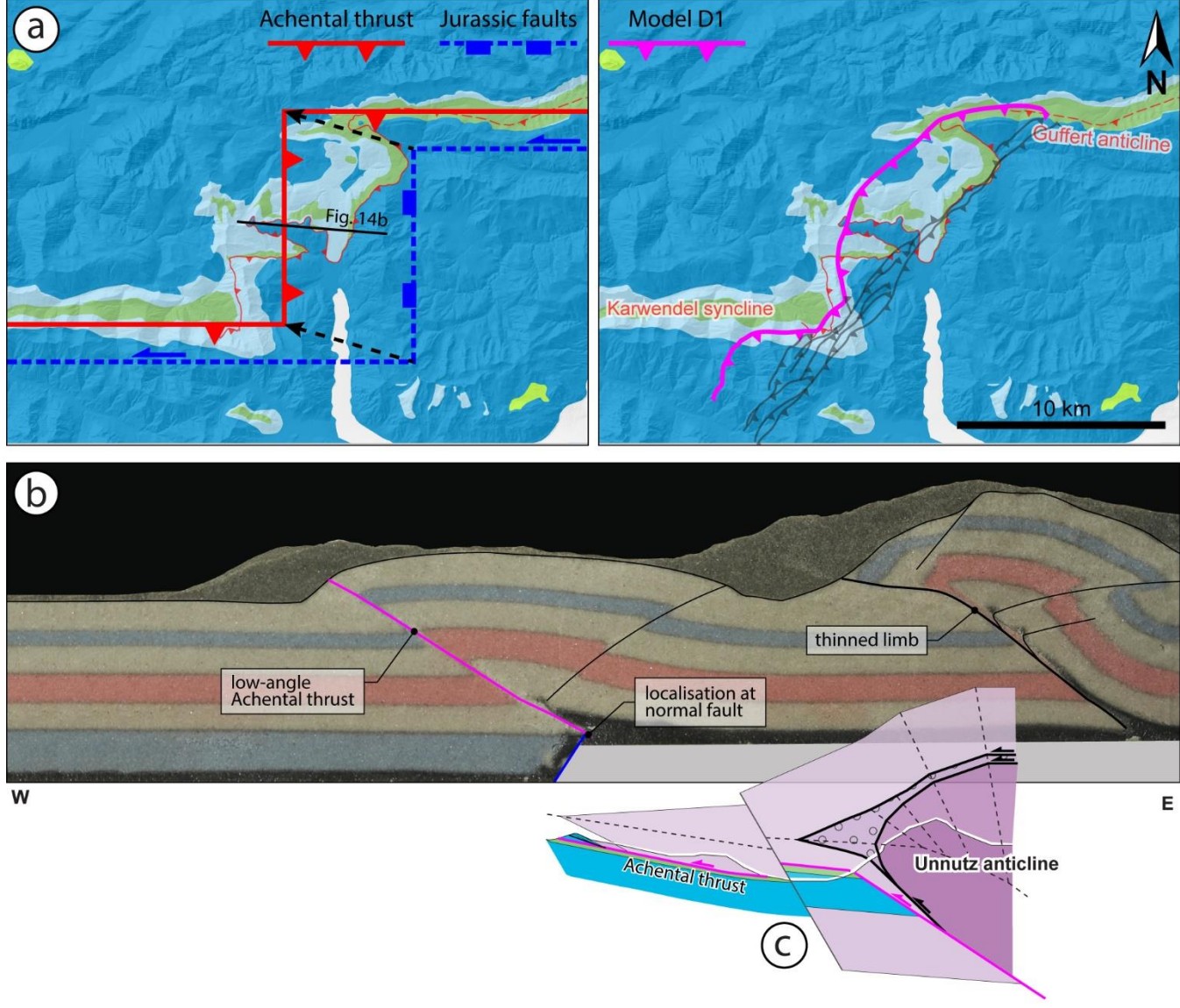

**Figure 14:** Summary figure showing (a) possible arrangement of Jurassic basin-bounding faults in relation to the Achental thrust (left) and overlay of inversion structures formed in analogue model D1, outlining the subsurface architecture (right). The trace of the thrust in model D1 has not been intersected with the topography. Geological map modified after Ortner and Gruber (2011) and Ortner and Kilian (2016). For legend of background figure, see Figure 1. (b) shows key structural elements in a cross-section of model D1 and (c) the corresponding cross-section of the Unnutz anticline (see also Figure 5a). The Achental thrust is marked in pink. See Figure 5 for legend of lithologies.

## 6 Conclusions

Analogue models were used to infer favourable kinematic and mechanic conditions for the oblique inversion of a pre-existing extensional basin margin. Yet relevant for understanding deformation geometries arising from oblique basin inversion in general, the modelling results have been discussed in the frame of the Achental structure in the European Alps, which evolved from the inversion of the Jurassic Achental basin during Cretaceous and Paleogene orogenic phases. The modelling set-up was based on the assumption that the Achental basin in the subsurface of the Achental structure consists of at least (1) an approximately N–S trending, W-dipping normal fault, and (2) two E–W trending additional strike-slip faults. Furthermore, we applied a thin-skinned style of deformation and shortening at 45° to the Jurassic structures, which both characterise Cretaceous orogeny in the NCA.

The modelling results show that the basal ductile décollement of the NCA (Haselgebirge-Reichenhall succession), which separated the basement from a predominantly brittle cover, is crucial for the decoupling of the sedimentary cover. The mechanical contrast across the basin bounding normal fault then controls deformation localisation at the existing fault step. As

a result, a low-angle thrust (Achental thrust), antithetic to the Jurassic normal fault, forms at the eastern margin of the Achental

basin. The hanging wall geometry of this thrust outlines the basin architecture in the subsurface, exemplifying the control of the pre-existing structural framework on structures that form during subsequent shortening. We therefore show that a single phase of Cretaceous oblique shortening at 45° to the Jurassic Achental basin axis is sufficient to form a sigmoidal geometry of the hanging wall (Achentaler Schubmasse). In a broader framework, we conclude that when shortening is applied oblique to existing basin boundaries, strain localisation leads to the formation of new thrusts along the steps formed by these structures.

**Author contributions**

The conceptualization of the study was done by HO and EW; Field investigations were performed by WvK, HO, AG and TS; Analogue modelling was planned and performed by WvK, EW and DS; Data preparation, analysis and visualisation were performed by WvK; WvK prepared the original draft of the manuscript with input from all co-authors; All co-authors reviewed and edited the manuscript and approved the final version.

**Competing interests**

At least one of the (co-)authors is a guest member of the editorial board of Solid Earth and has collaborated as a (co-)author on publications involving one of the guest editors of Solid Earth. The peer-review process was guided by an independent editor, and the authors have also no other competing interests to declare.

**Acknowledgements**

This work was completed as part of a Master Thesis at the University of Innsbruck. The University of Innsbruck through a needs-based scholarship provided financial support covering fieldwork and travel expenses. Laboratory experiments were conducted at the TecLab of Utrecht University. Klaus Pelz and an anonymous reviewer have provided valuable feedback, which has considerably improved the quality of this work. We sincerely thank Antoine Auzemery for assistance with the conceptualization and realization of analogue experiments, and Taco Broerse for his help with the Strainmap package, as well

as fruitful discussions. We also thank Lukas Schifferle for assistance during fieldwork and digitalization. The authors thank Midland Valley for providing their Move Software in the frame of their academic software initiative and the state of Tyrol for providing high resolution hillshades.

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
