# Peer review of "Fold localisation at pre-existing normal faults: Field observations and analogue modelling of the Achental structure, Northern Calcareous Alps, Austria"

_EGUsphere, 2022_

## Author Response (AR1)

Innsbruck, July 31, 2023

**Revised manuscript for publication in Solid Earth**

Dear editor,

We would like to thank you for giving us the opportunity to revise our article "**Fold localisation at pre-existing normal faults: Field observations and analogue modelling of the Achental structure, Northern Calcareous Alps, Austria**" (revised title) by Willemijn S. M. T. van Kooten, Hugo Ortner, Ernst Willingshofer, Dimitrios Sokoutis, Alfred Gruber and Thomas Sausgruber.

We are glad to have received such detailed and constructive feedback from the reviewers that helped us to improve our manuscript and we would like to thank them for their effort. In the revised version of the manuscript we incorporate all the suggestions of the reviewers. More specifically, we have striven to reduce the length and complexity of the manuscript, and to improve its readability in particular for those not familiar with the study area.

Below we respond to all the concerns raised by the reviewers and reference the edits in the revised manuscript.

Thank for your consideration of our revised manuscript.

On behalf of the authors,

Willemijn van Kooten

Willemijn van Kooten, MSc
Teaching & Research Assistant
Digital Business & Software Engineering (BSc)
Tel: +43 512 2070-4322, Fax: -1099
willemijn.vankooten@mci.edu

**Reviewer 1 - Anonymous**

The article entitled "Oblique basin inversion leads to fold localization at bounding faults: Analogue modelling of the Achental structure, Northern Calcareous Alps, Austria" submitted by Willemijn and co-authors is well written and with figures in a good style but it is also very long and a bit hard to follow. This is in part due to the problems I had trying to link what the overwhelming text was explaining and the figures were indicating. These last statements, however, do not question the validity of the study or the results from analogical modelling. The main objective of the work seemed to me to find the potential best solution for the structure at depth to propose solutions that better match one of these interpretations (sometimes controversial) for the hidden structure underneath the Achental structure, analysed by means of fieldwork and cross-section construction previously by (reference).

We thank the reviewer for the constructive feedback. In the revised manuscript, we have incorporated the feedback, reduced the manuscript length and increased its readability for a broader audience by improving on clarity of figures and text.

Geological Setting is a description at very large-scale long-lasting evolution of events from Pangea margins at the end of the Permian to the closure of the Tethys during the Paleogene of the study region passing through the significant phase for these works that it is the Jurassic extension that affected the Alps and all without any figure to help follow it (especially for those who do not know the region first hand.

Point taken. For increasing the readability of the geological setting section, we have included a figure (Figure 2) that summarizes the tectonic evolution of the Northern Calcareous Alps and provides visual support to the text in the manuscript.

Line 52 & Figure 1: Figure 1 showing the very schematic geology of the Achental structure on a geological map that corresponds to a small box located on a small map (inset) of Austria where the Northern Calcareous Alps have been represented in black. I think that a simplified geological map of the Northern Calcareous Alps would be indispensable to understand the tectonic context in which the Achental structure is located. Figure 1b) is fine but perhaps a simple cross-section crossing the thrust zone to show the thickness distribution of the Upper Jurassic Oberalm Fm. on both blocks of the Achental thrust would also help to constrain the potential geometry of extensional fault based on thickness variations.

We have revised Figure 1 to include a simplified tectonic map and cross-section of the Northern Calcareous Alps.

Line 100 & Figure 2: The description of the stratigraphy is exhaustive and complex. Figure 2 is difficult for me to understand because of the colours used, due to the lack of lithological patterns within the colours and the lack of sedimentary thicknesses. The colours do not follow the specific colours for the different ages typical and recommended from the International Chronostratigraphic Charts. The lithologies are not shown and the rheology of the sedimentary succession in not shown. Nor is the thickness of the different sedimentary successions (you could even add minimum to maximum thicknesses) if it is very variable. And an important issue is the lack of associated rheology since the shape of the stratigraphic table does not show the hardness or weakness of the material that is very important for the mechanical stratigraphy.

Also, together with the mechanical stratigraphy, the potential intermediate detachment levels could be added (evaporites, shales, etc…) making a better link with the cross sections and the analogue models.

In addition, in the cross-sections of Figure 4 there are only 5 different (colours) successions and in the model set up there are only 4-5 stratigraphic units. Then, do you need this super complex stratigraphic table in Figure 2 where no direct mention is made regarding the units that are important to explain the cross-section and the model set-ups?

We thank the reviewer for these valuable comments, which we have implemented as follows: 1) we have shortened and simplified the description of the stratigraphy (**LINES 101–152**); 2) we have furthermore revised Figure 2 (**Figure 3** revised manuscript) using ICC-recommended colors and USGS lithological patterns, and have included sediment thickness. We have furthermore modified the stratigraphic column to address the reviewer's concern that intermediate detachment levels and rheology are not shown, and have improved the connection of the stratigraphy to the cross sections and the model set-ups by using identical colors and designations.

Line 165 & figure 3: I already understand that the map and the cross sections come from previous works and have not been modified in this study but it is very difficult for me to understand the geological-tectonic map (although it is very beautiful and well done) because everything is a bit "like painted with pastel colours" and with the main lines of the thrusts very thin. The names (labels) of the different anticlines and thrust are so small that are difficult to find them.

We have also included ICC-recommended colors in the geological map and cross-sections to correlate with the stratigraphic chart in **Figure 3** (revised manuscript). Furthermore, we have revised the geological map **Figure 4** (revised manuscript) for better readability (e.g., label size and opacity have been increased).

Line 165 & Figure 4 / 5: The cross sections, which have already been published previously and which match the geological map, need the names of the main overlaps (labels) to make them more understandable. In any case, the cross-sections do not show or allow to observe the tectonic inversion that is proposed to be modelled on this paper. This is an inconvenience together with the fact (already explained in the paper) that the fault thrust has an opposite dip to that of the supposed normal fault of the model. Although I am not discussing the presented geological cross sections, the large overturned anticline (largely eroded) in Figure 4B shows a culmination of the anticline that is almost 4 km in length (?) although it is not very significant for the results of the paper.

We have added labels to the cross-sections, which correspond to the labels on the geologic map. We agree with the reviewer that it would be good for the tectonic inversion to be visible in the cross-sections. However, in the absence of suitable seismic information, it is not possible to create a reliable model by extending the cross-sections in depth based on surface geology only. The length of the Guffert anticline culmination can be explained as follows: the axial surface of the Guffert anticline (Figure 5b in revised manuscript) in fact has two culminations, which result from the refolding of the Achental thrust and its hanging wall above a deeper décollement. The fold axis in this part of the Unnutz mountain is parallel to the section plane and thus shows an extreme length in Figure 5b. Please also compare Figure 5a (revised manuscript), where the trace of section 5b is indicated.

I am also curious on the fact that this large region of Calcareous Alps is beginning to be re-interpreted with an important component of salt tectonic while this document does not refer to it (very shortly in the lithostratigraphic section). Did you have considered incorporating some model involving large overturned domains as recently done in other re-discovered regions in the Tethyan fold belts where halokinetic processes are important and simplify some previous complicated structural interpretations?

Point taken. We have added text to the geologic setting and the discussion (**LINES 120–123; 583–594; 627–647**) to include the ongoing discussion on salt-related structures in the Northern Calcareous Alps.

Line 265 and Figure 5: It is not very clear to me what criteria are used to justify the geological set up presented in Figure 5 if it is not supported by the map and geological sections presented as a support and starting point for the analogical models... I also suggest that the models A to D2 (in text and Figure 6 and subsequent ones) could be labelled in addition with short label illustrating each of the models by their characteristic structure or / and characteristic stratigraphy). Otherwise it becomes difficult to remember what are their main parameters throughout the description of each of the models in the text.

We thank the reviewer for bringing this up. We have clarified the points addressed by the reviewer by 1) updating **Figure 1** to include more applicable maps and cross-sections, 2) revising the stratigraphic column (**Figure 3**) to match **Figure 6** (revised manuscript), and 3) extending the reasoning and evidence for the Jurassic basin geometry in Section 2.2 (**LINES 225–240**). The proposed Jurassic structure and Permian-Cretaceous (mechanical) stratigraphy form the framework for the analogue modelling. As stated in Section 3.4, creating a modelling basis requires simplification (**LINES 352–353**). Therefore, the analogue models are not able to represent all the intricacies of the natural example. We have included this in Section 3.4 "Limitations and simplifications" (e.g., **LINES 367–370**) and in Section 5 "Discussion" (**subsections 5.2.1 and 5.3**). To address the second part of this comment and improve readability, we have included key model parameters in all figures in Section 4 "Results" (**Figures 8 to 13**).

Line 535 and Figure 13: Apart from the detailed description of each of the models which is long and very descriptive, in the Discussion chapter there is another very long part of the paper on analogue modelling results (pages 24 to 26) and then another section as well long (page 26 to 28) in which models are compared with the example from nature but only on a map (in plan). The comparison of the resulting cross sections of the models with the geological cross sections of Figure 4 do not show many similarities apart from perhaps C Figure 9, model B of Figure 10, and A of Figure 12 but none of these models present inverted flanks neither in the footwall nor in the hanging wall of the east-dipping main thrust.

We concur that the model descriptions are somewhat lengthy in some instances. In the revised version of the manuscript we have attempted to shorten the model descriptions in **Section 4**. Furthermore, we have rewritten the discussion (in particular **LINES 547–655**) and updated **Figure 14** (revised manuscript) to compare the models to the natural example both in map and cross-sectional view. With reference to the lack of inverted flanks in our experimental results, we remark that correctly scaling the mechanical stratigraphy of the natural example for length results in relatively thin viscous layers and a style of deformation that is controlled by the brittle layers. As such, quartz sand, the modelling material simulating brittle behavior, is not able to reproduce the inverted flanks as seen in the cross-sections. We acknowledge this limitation, which we have discussed in Section 5 of the revised manuscript (**LINES 563–582**). We could have chosen for an approach in which we tune the modelling setup to obtain a "look-alike" result, by for example introducing unproportionally thick viscous layers, but this never was the purpose of this study. Our main task was to better understand strain localization and associated deformation by folding and thrusting at oblique basin bounding faults, which we successfully completed.

Conclusions: The conclusions' two first paragraphs only indicate the starting points for modelling. In addition, data presented only show indirect constraints that are used since neither the map nor the cross-sections show any inverted basin. The other conclusions would be valid if the starting points were very solid. In this case in which the data only permits to prepare assumptions (which may be true), the analogue model indicates that a geometry similar to the one mapped on the surface can be achieved with a single oblique compression along the N-S normal fault trace, but do not match very well the structure illustrated by the geological cross sections...

To arrive at conclusions that are supported by proper arguments several measures have been taken throughout the manuscript. We have rewritten and revised parts of the manuscript (e.g., **Section 2.2, 3.1, 3.2, 5**) to clearer present the arguments for the existence and architecture of a Jurassic basin, which was used as a starting point for modelling. Furthermore, we have improved on the comparison between the deep structure, predicted in the analogue models and that of the natural example. These measures prepare the reader better for the conclusions (**Section 6**) where we have now clearly distinguished starting points from conclusions. Again, we would like to point out that the simplifications in the analogue modelling restrict a comparison to the first order features. As such, some geologic features of the natural example (shown in geological map and cross-sections) may not be represented adequately in the analogue model. Increasing complexity within the analogue models may increase the likeness between model sections and geological cross-sections, but clouds the main task set for the modelling: to analyze deformation localization at oblique basin boundaries.

As a summary, the methodology used in the paper is correct and it is well written and with very stylish figures, but the article is long and heavy (and it took me a lot of work to go through the detail of the paper). I think that these shortcomings that I have found can be fixed to make the paper clearer and probably shorter and with better comprehensive figures so that a wider number of readers will find it easier to read and understand the difficulties and advantages of the work done to solve this important structural feature in the Northern Calcareous Alps.

We thank the reviewer for the time invested to provide this constructive feedback, which greatly helped to improve the quality of the manuscript.

**Reviewer 2 – Klaus Pelz**

The manuscript "Oblique basin inversion leads to fold localisation at bounding faults: Analogue modelling of the Achental structure, Northern Calcareous Alps, Austria" handed in by van Kooten et al addresses a very peculiar tectonic element of the Western NCA that plays a crucial role in defining structural styles in the tectonic evolution of the NCA. Using analogue models to understand the influence of inherited elements on structures we have looked at for some time and still struggle to explain consistently is a very efficient way to narrow down the range of valid interpretations. The paper frames the geological setting in a very concise way and clearly defines the boundary conditions for the different setups of the analogue models. The results of the analogue models are well presented and discussed with regards to the natural example. Good quality figures do illustrate all main points and help bridge back from experiments to nature.

The long scientific record on the Achental structure under investigation and its sigmoidal shape is summarized in a very condensed way that includes stratigraphic successions and thicknesses for the NCA, main tectonic elements, mechanical stratigraphy of involved units, and different interpretations and models on deformation phases, timing and indications for shortening directions. Also, the role of Jurassic extension and the resulting fault trends is discussed as main ingredients for the modelling.

Some minor issues I would appreciate to be addressed are:

the lack of an overview cross section that clearly shows the position of the Achental structure within the NCA thrust sheet systems. This would help better assess the relevance and the impact of the Achental structure compared to elements that accommodate the vast amount of shortening within the thrust system.

The Achental thrust trace in Fig. 1b does locally (around Achenkirch) not follow the geological overview map, which makes it harder to grasp where this map sits in relation to the overview map when first looking at it – a very minor issue.

The Eben thrust brings Haselgebirge-Reichenhall succession to the surface in Achensee region, but is not shown on a geological map. It might help to show the full stratigraphic range of the thrust sheets down to their base on a map, to better assess the scale of the Achental structure (might be addressed together with the first point).

We thank the reviewer for the feedback. We have revised **Figure 1** to include a simplified cross-section of the Northern Calcareous Alps. In this cross-section the Eben thrust (third comment) is also included. Furthermore, we have adjusted the trace of the Achental thrust in **Figure 1c** to achieve consistency with the geological map (second comment).

Haselgebirge-Reichenhall fms (line 110) and Raibl fm (l 128) are clearly highlighted as incompetent parts of the pile, but Raibl is not shown as such in Fig. 5.

The Raibl Fm is not modeled in all experiments as incompetent layer. This depends on the specific modelling set-up. We have included this information in **Figure 6** (revised manuscript) and in e.g., **LINES 130–132, 264–266**. For the final models we have decided to model the Raibl Fm as a competent unit, to simplify the modelling set-up.

Salt tectonics is mentioned in (l 122) as being responsible for thickness variations in the Triassic, but not further elaborated on – see further below.

Following the reviewer's suggestion, we have elaborated on this in Section 2.2 (**LINES 120–123)** and have included a discussion on salt tectonics in Section 5 (**LINES 583–594; 627–647**).

Stating that the SE dip of the thrust is strongly opposed to the steep E- to SE dip of the strata, ruling out a stratigraphic contact between these formation (l 180) needs some clarification in my opinion as it indicates that the displacement is significant although on map view displacement looks minimal.

We agree with the reviewer that this needs clarification. The thrust dips SE, and cuts across the hinge of the Seebergspitze syncline, which is offset. North of the Seebergspitze syncline hinge, the strata dip E; west of the hinge the strata dip vertical to S. In this area, the thrust is out-of-sequence with respect to the Seebergspitze syncline, but the offset is minor and dies out not much further to the west. We have clarified this in the text in Section 2.2 (**LINES 168–173**).

The Scharfreuter syncline in the text (l 182) is actually an anticline on the map

It is indeed an anticline. We have also corrected this in Section 2.2.

Also, the Montscheinspitze as part of the of the Guffert-Unnutz-Montscheinspitze anticline fold train (l 200) is not displayed on a map.

We have included the Montscheinspitze anticline in the geological map in **Figure 4**.

For the modelling part, I think it would be helpful to clearly describe why certain setups were chosen. For example, it is not evident why certain models do have silicone both on the table and on the elevated base plate and some only on the elevated base plate. What is the rationale for having silicon only on the elevated basement block? Also, why certain experiments were conducted with a mobile base plate and some were fixed, not clearly relates to distinct and different assumptions that are communicated. And if certain models, e.g. model C, are run using fixed base plate. A careful description of what has been chosen and why would improve this part of the MS, and probably a table summarizing it – or adding this info to Fig. 6.

In the revised version of the manuscript, we added details on the mechanical and kinematic variability of the different models and their justification (**Section 3.2; LINES 294–328**). Furthermore, we follow the reviewer's suggestion and provide a new table (**Table 2**) summarizing modelling parameters, which benefits the readability of the manuscript.

In l 334, a simple fault arrangement for the setup is described with one N-S and two E-W striking elements and NW directed shortening, which is in contradiction to models A, B and C, where shortening is W-directed, and only valid for models D. This is not clarified in the text. Only models D1 and D2 are shortened oblique to pre-existing trends. Why?

Like in other analogue (or numerical) modelling studies a series of experiments has been designed which starts with some basic models to understand the deformational response to a set of relatively simple geometric, kinematic and rheologic initial conditions. In this context, Models A, B and C are shortened perpendicular to the basal plate to investigate favorable rheological parameters for the inversion phase. Thereafter, we decided to progress towards more complex models including oblique shortening. In the revised version of the manuscript, we have clarified the modelling strategy in **Section 3.1 and 3.2**.

In Figure 12b (D1, NE-SW section), the edge of the basement/basal plate is unclear - is it tapered?

The tapered appearance of the basal plate is due to the continuation of the plate cutting into the image in an inconvenient way. We have edited **Figure 13** (revised manuscript) to resolve this.

When reviewing the individual experiments, it would be helpful to have a small inset that shows the simplified initial setup of the experiment, especially useful for figs 9 and 12. This would facilitate relating styles and structures to configurations when reading through the manuscript, or even browsing the figures.

We have included a small inset with modelling parameters in all figures in Section 4 (**Figures 8 to 13**).

What is then deduced convincingly in the discussion is that no multiphase deformation is needed for the observed geometry: The analogue model can explain the sigmoidal shaped geometry of the HW and the deformation within a single phase of shortening. The complexity of the analogue model, however, only partly can explain the natural example: A striking first order feature of the discussed area is the existence of large overturned panels of Triassic successions, best illustrated with two sections in Fig 4.

All assessed thrust displacements cited in the MS are based on cut-offs from exactly these inverted panels (Natterwand, Plickenkopf): Along the main E-W trending Thiersee and Karwendel synclines, apparently only minor offset to blind thrusting remains to be measured along the thrust trace/tip. For the N-S section, 5-7 km are assessed (Fig. 4) based on inverted panels, but no displacement is indicated beneath overturned Natterwand on the map. In the E-W section, the thrust contact is much clearer (fully inverted Upper Triassic in the HW on Lower Cretaceous, but very low cut-off angle) with published 8 km of minimum offset. Further to the SW, again thrust displacement appears to vanish along the E-W trending Karwendel syncline.

So highest displacement is recorded exactly at the NE-SW overstep between these major E-W trending synclines, and overturned panels are dominant exactly there: this is not explained by the results of the analogue model. Furthermore, the process that forms these huge inverted panels highly impacts the amount of shortening we are calculating: The rollover-fault-prop model suggested by Storti & Salvini in 1996 does have its limitations when applied to natural examples. Any tilting that might have occurred and contributes to the rotation of these panels prior to shortening needs to be put into perspective.

For the Achental thrust, the best estimate of 5-7 km offset is measured from the N-S section in Figure 5b (revised manuscript), based on the cutoff points of the Jurassic strata. Since the footwall cutoffs are exposed neither in the N-S, nor the E-W section, displacements are not exactly constrained, although they do not differ much between both sections. We agree with the reviewer that specific features of the natural example and sections, the inverted panels in particular, cannot or only partially be explained by the analogue models. This is largely the result of the chosen modelling materials and scaling. In the revised version of the manuscript (**Section 5 "Discussion"; LINES 547–655**) we highlight these differences and discuss underlying causes. Although the rollover-fault-propagation model by Storti and Salvini (1996) may have its limitations, one of those being that it does not explain the hinge collapse as interpreted in Figure 5a (revised manuscript), this model does produce overturned panels. Even if salt tectonics should play a role, as suggested by the reviewer, there are other models than the rollover-fault-propagation model that also produce overturned panels (see e.g., Wallace & Homza, 2004).

Given the recent indications for salt tectonics from more and more areas within the NCA and taking into consideration the range of structures and especially tilting of beds during salt evacuation, this could change the view on processes behind controlling Achental structure. Analogue models do look much different when not applying layer cake geometries pre-shortening. Basement faults that control

distribution of salt have the potential to define mini-basins, their extent, and their boundaries. And have local effects when shortened that might look pretty similar to Achental structure. At least discussing the impact and consequences of non-layer-cake starting geometries would enhance the significance of this MS.

Point taken. We recall that the primary goal of our modelling study was to gain insight in conditions controlling strain localization and associated folding and thrusting at pre-existing basin boundaries in an oblique convergent setting (see also **LINES 637–639**). From this perspective the models demonstrate, as acknowledged by the reviewer, that only a single phase of deformation is needed to explain key elements and kinematics of the natural example. Increasing the amount of silicone putty and changing its properties, would have been required to properly scale with the dynamics of salt tectonics, which was not the aim of this study. While interesting from a modelling point of view, this would create flaps when including Triassic salt dynamics. We did not aim for this complexity. However, in the revised version of the manuscript, we acknowledge that non-layer-cake starting geometries would impact on the modelling results. We follow the reviewer's suggestion and included this aspect in the discussion (**LINES 627–647**).

We thank the reviewer again for the time invested to provide this constructive feedback, which greatly helped to improve the quality of the manuscript.

**References**

Storti, F., & Salvini, F. (1996). Progressive Rollover Fault-Propagation Folding: A Possible Kinematic Mechanism to Generate Regional-Scale Recumbent Folds in Shallow Foreland Belts. AAPG Bulletin, 80. https://doi.org/10.1306/64ED8782-1724-11D7-8645000102C1865D

Wallace, W. K., & Homza, T. X. (2004). Detachment folds versus fault-propagation folds, and their truncation by thrust faults. AAPG Memoir, 82, 324–355.

---

## Author Response (AR2)

Innsbruck, September 15, 2023

**Revised manuscript for publication in Solid Earth**

Dear Professor Schreurs,

We would like to thank you for your comments and for giving us the opportunity to improve our article "**Fold localisation at pre-existing normal faults: Field observations and analogue modelling of the Achental structure, Northern Calcareous Alps, Austria**" by Willemijn S. M. T. van Kooten, Hugo Ortner, Ernst Willingshofer, Dimitrios Sokoutis, Alfred Gruber and Thomas Sausgruber. We have incorporated all suggestions and are glad to present a final revised version of the manuscript.

Below we respond to all the suggestions and reference the edits in the revised manuscript.

On behalf of the authors,

Willemijn van Kooten

Willemijn van Kooten, MSc
Teaching & Research Assistant
Digital Business & Software Engineering (BSc)
Tel: +43 512 2070-4322, Fax: -1099
willemijn.vankooten@mci.edu

**Topic Editor:**

Fig. 1. When printed the blueish colors all merge into one single "blue" both in the overview map and cross-section. The different units can no longer be distinguished. Please improve, e.g. by adding a pattern (dots or symbols) to the various units

We thank the editor for this excellent suggestion and have improved the graphical presentation of the various units.

Fig 1 (a) should be: Tectonic map showing the location of the Northern Calcareous Alps in Austria......

Figure caption changed to: "Figure 1: (a) Tectonic map showing the location of the Northern Calcareous Alps in western Austria. […] (c) Geological map of the Achental structure within the Karwendel thrust sheet. Black lines show isopachs of the Upper Jurassic Oberalm Fm […].

Throughout the text: it should be "Rhaetian" instead of "Rhätian"

Done.

In Table 1: unit of strain is "1/s" (and not m/s)

Strain rate in Table 1 changed to "Convergence rate".

In Table 2: Plate border designation should be "Straight" "instead of "Straigth"

Done.

Line 415: "analogue model" instead of "analoging"

Done.

In all figures presenting model surface results, I suggest that the figure panels reflect shortening increasing from left to right; and that the individual model results (of one particular experiment) are always presented in rows.

We have implemented these changes in all figures presenting model surface results.

Line 537: "inversion" instead of "inversional"

Done.

Further changes in the revised manuscript include improvements in Figure 2, small corrections in figures, tables, spelling and grammar, and updating the citation style to match the journal requirements.

---

## Author Response (AR3)

Innsbruck, October 18, 2023

**Revised manuscript for publication in Solid Earth**

Dear Professor Schreurs,

We would like to thank you for your comments and for giving us the opportunity to improve our article "**Fold localisation at pre-existing normal faults: Field observations and analogue modelling of the Achental structure, Northern Calcareous Alps, Austria**" by Willemijn S. M. T. van Kooten, Hugo Ortner, Ernst Willingshofer, Dimitrios Sokoutis, Alfred Gruber and Thomas Sausgruber. We have incorporated all suggestions that had not been incorporated yet in the last round of revisions and are glad to present a revised version of the manuscript.

Below we respond to all the suggestions and reference the edits in the revised manuscript.

On behalf of the authors,

Willemijn van Kooten

Willemijn van Kooten, MSc
Teaching & Research Assistant
Digital Business & Software Engineering (BSc)
Tel: +43 512 2070-4322, Fax: -1099
willemijn.vankooten@mci.edu

**Topic Editor:**

Modify the caption of Fig. 1a as it is a map showing the main tectonic units of W-Austria, with the location of the study area shown in Fig. 1c.

Done.

Line 113: "Fig. 2" should be "Fig. 3"

Fig. 2 refers to the figure in Spötl (1989). We have changed the text to "their Fig. 2".

Line 118: "Fig 1" should be "Fig. 1a,b"

Done.

Line 159: use "Their Fig. 15" if this figure refers to the publication of Brander & Gruber, 2011).

Done.

Add to caption of Fig. 3 what the thicknesses on the right-hand side mean (these are not the thicknesses used in the models), and why the "Raibl beds can be considered either as "competent" or "incompetent". Make the link to the analogue models in the caption.

The thicknesses on the right-hand side of Fig. 3 correspond to the thicknesses in Fig. 6 (see revised manuscript). We have referred to the layers 1–4 that are depicted in both Fig. 3 and 6 and have altered the caption of Fig. 3 to:

"Figure 3: Stratigraphic overview of the Permian–Cretaceous sedimentary succession of the Karwendel thrust sheet in the Achensee area, modified after Kilian et al. (2021). Various sedimentary strata have been assigned to different incompetent and competent layers (1–4) in the analogue model set-up (see also Fig. 6). The Raibl beds were modelled as incompetent or competent, depending on the specific model. Kilian et al. (2021) discuss the rheological behaviour of the Raibl beds in more detail."

Caption of Fig. 6. Leave out "inversional"

Done.

Line 333: place the table caption below Table 1.

Done.

Line 375: place the table caption below Table 2.

Done.

Fig. 11. I would interchange (b) and (c), such that the cumulative strain results are directly below the corresponding surface photographs. But also in this figure, I would show increasing shortening from left to right.

Done.

Throughout the manuscript text use "Fig." instead of "Figure", except at the start of each figure caption

Done.

When printed the bluish colors in Fig 1a and Fig 1b all merge into one single "blue", and the different tectonic units are not distinguishable. Please add a pattern (e.g. dots, triangles, etc.) to distinguish the various units.

Figure 2. Please label the three panels, (a), (b) and (c), and refer to them accordingly in the figure caption when describing the tectonic evolution. Right now, there is a (d).

Throughout the manuscript, when referring to the uppermost stage of the Triassic Period, it should be "Rhaetian" instead of "Rhätian". Also change in Fig. 3 and 5.

In the experimental figures, I would place the panels from one single experiment in a row, (as for example is done in Fig. 11), and showing increasing shortening from left to right.

In fig. 8, turn the cross-section clockwise by 90° and place it below the four panels (place the four panels in one row, as suggested above) with increasing shortening from left to right.

These comments have been addressed in the previous round of revisions.